# *Caenorhabditis elegans* male sensory-motor neurons and dopaminergic support cells couple ejaculation and post-ejaculatory behaviors

Brigitte LeBoeuf, Paola Correa, Changhoon Jee, L René García*

Department of Biology, Howard Hughes Medical Institute, Texas A&M University, College Station, United States

**Abstract** The circuit structure and function underlying post-coital male behaviors remain poorly understood. Using mutant analysis, laser ablation, optogenetics, and $Ca^{2+}$ imaging, we observed that following *C. elegans* male copulation, the duration of post-coital lethargy is coupled to cellular events involved in ejaculation. We show that the SPV and SPD spicule-associated sensory neurons and the spicule socket neuronal support cells function with intromission circuit components, including the cholinergic SPC and PCB and the glutamatergic PCA sensory-motor neurons, to coordinate sex muscle contractions with initiation and continuation of sperm movement. Our observations suggest that the SPV and SPD and their associated dopamine-containing socket cells sense the intrauterine environment through cellular endings exposed at the spicule tips and regulate both sperm release into the hermaphrodite and the recovery from post-coital lethargy.

## Introduction

Persistence in performing a goal-orientated behavior must be balanced by behavioral termination cues once the task is completed. One such behavior, mating, is important for species propagation and can improve an individual's ability to cope with stress (*Neumann, 2009*). In humans, rats, and other animals, a period of disinterest and mating inability follows ejaculation in males (*Beach and Holz-Tucker, 1949*; *Masters and Johnson, 1966*; *Barfield and Geyer, 1972*; *Oomura et al., 1983*; *Ureshi et al., 2002*). Primarily studied in rodents, sexual disinterest and inability following mating are described in two ways: the refractory period, defined by the short term duration between consecutive ejaculations (*Levin, 2009*), and sexual satiation or exhaustion, a period of time following repeated copulations where the male rats require 6–14 days to regain sexual potency (*Beach and Jordan, 1956*). If a male rat is considered to be sexually satiated, he cannot sire progeny even if he engages in copulatory activity (*Tlachi-Lopez et al., 2012*; *Lucio et al., 2014*).

While the behavioral phenomenon has been described, little is understood about the molecular and cellular mechanisms controlling both satiation and the refractory period. Neurotransmitters and hormones such as serotonin and prolactin may extend the period of inactivity, while others such as dopamine and norepinephrine may shorten it (*McIntosh and Barfield, 1984a*, *1984b*, *1984c*; *Buvat et al., 1985*; *Marson and McKenna, 1992*). However, the basic structure and function of mating circuits that exhibit a period of inactivity are still being elucidated (*Levin, 2009*; *Turley and Rowland, 2013*).

The well-defined structural components of the nervous system in the *Caenorhabditis elegans* hermaphrodite have facilitated a detailed understanding of how circuits function to produce behaviors (*White et al., 1986*; *Varshney et al., 2011*; *Cohen and Sanders, 2014*). Combining the anatomical information with optogenetics, cell ablations and calcium imaging have uncovered information on how *C. elegans* responds to both attractive and repulsive stimuli (*Cohen and Sanders, 2014*). For example,

*For correspondence: rgarcia@
bio.tamu.edu

Competing interests: The authors declare that no competing interests exist.

**eLife digest** The nematode worm, *C. elegans*, is roughly 1 mm long, made up of around 1000 cells and has two sexes: male and hermaphrodite. Hermaphrodite worms produce both eggs and sperm and can self-fertilize to generate around 300 offspring each time. Fertilization by a male, on the other hand, results in three times as many progeny and introduces genetic diversity into the population. However, it also reduces the lifespan of the hermaphrodite.

Mating also incurs a cost for males: it requires a lot of energy, which prevents male works from engaging in other activities, such as feeding, and it also increases their risk of predation. In many species, including *C. elegans*, the frequency with which a male can mate is limited by a period of reduced mating drive and ability that follows each instance of successful mating. However, the molecular and cellular basis of this 'refractory period' remains largely unclear.

Using a range of techniques, LeBoeuf et al. have now identified the circuits that regulate male mating behavior in *C. elegans*. When male worms were introduced into a Petri dish containing 15 hermaphrodites, most males initiated mating within about 2 min. The length of the refractory period varied between worms, but averaged roughly 12 min. This consisted of a period of disinterest, in which males did not approach hermaphrodites, followed by a period in which males attempted mating but were slower and less efficient, suggesting that the neural circuits controlling mating behaviors had yet to recover completely.

Males with longer refractory periods produced more progeny in their second mating than those with shorter refractory periods, suggesting that the interval also enables males to replenish their sperm levels. Further experiments revealed that a chemical transmitter called dopamine promotes ejaculation and then immediately reduces the worm's activity levels, giving rise to the refractory period.

By enforcing a delay between matings, the refractory period may also increase the likelihood that successive matings will be with different hermaphrodites, helping to maximize the number and diversity of offspring. Some aspects of the neural circuitry that controls the refractory period in *C. elegans* resemble those seen in mammals, suggesting that insights gained from an animal with 1000 cells could also be relevant to more complex species.

*Li et al. (2011)* identified the sensory neurons and their direct downstream targets that regulate response to the noxious stimuli of a harsh touch (*Li et al., 2011*). *Hendricks et al. (2012)* determined which neurons controlled head movement in response to the chemo attractant isoamyl alcohol (*Hendricks et al., 2012*). Additionally, several studies highlight the role that extrasynaptic neuro-modulation plays in regulating behavioral responses, adding another layer to neuromuscular circuit control of behavior (*Flavell et al., 2013*; *Leinwand and Chalasani, 2013*). The tool set used to decon-struct the circuits in hermaphrodites can be applied to study the most complex behavior exhibited by the nematode, male mating.

Previous work on the mating steps that precede ejaculation provides a foundation for under-standing the circuit structure and function that produces copulation-induced inactivity. Reconstruction of serial electron microscopy images provides detailed information of the structure and connectivity of the male tail that is not available in other species (*Sulston et al., 1980*; *Jarrell et al., 2012*). The con-nectome has then been utilized as a tool to determine how circuits allow the flexibility necessary for executing a multi-step goal-oriented behavior. We and others have undertaken multiple studies to elucidate how the connectome functions to produce male mating (*Liu and Sternberg, 1995*; *Barr and Sternberg, 1999*; *Hurd et al., 2010*; *Wang et al., 2010*; *Koo et al., 2011*; *Miller and Portman, 2011*; *Siehr et al., 2011*; *Barrios et al., 2012*; *Garrison et al., 2012*; *Sherlekar et al., 2013*).

*C. elegans* males intromit by initially prodding the tightly closed hermaphrodite vulva slit with their two copulatory spicules (*Figure 1A,B*). After the spicules breach the vulval slit and fully penetrate, the males transfer sperm (*Figure 1B*; *Liu and Sternberg, 1995*). Coupling proper spicule position with prodding is coordinated via cholinergic and glutamatergic signaling from the left-right bilateral post cloacal sensilla (p.c.s.) (*Figure 1C*). These neurons sense the vulva using sensory processes that project posteriorly from the cloacal opening. They stimulate the sex-specific oblique and gubernaculum mus-cles that, via gap junctions, induce twitch contractions in the spicule protractor muscles (*Figure 1C*).

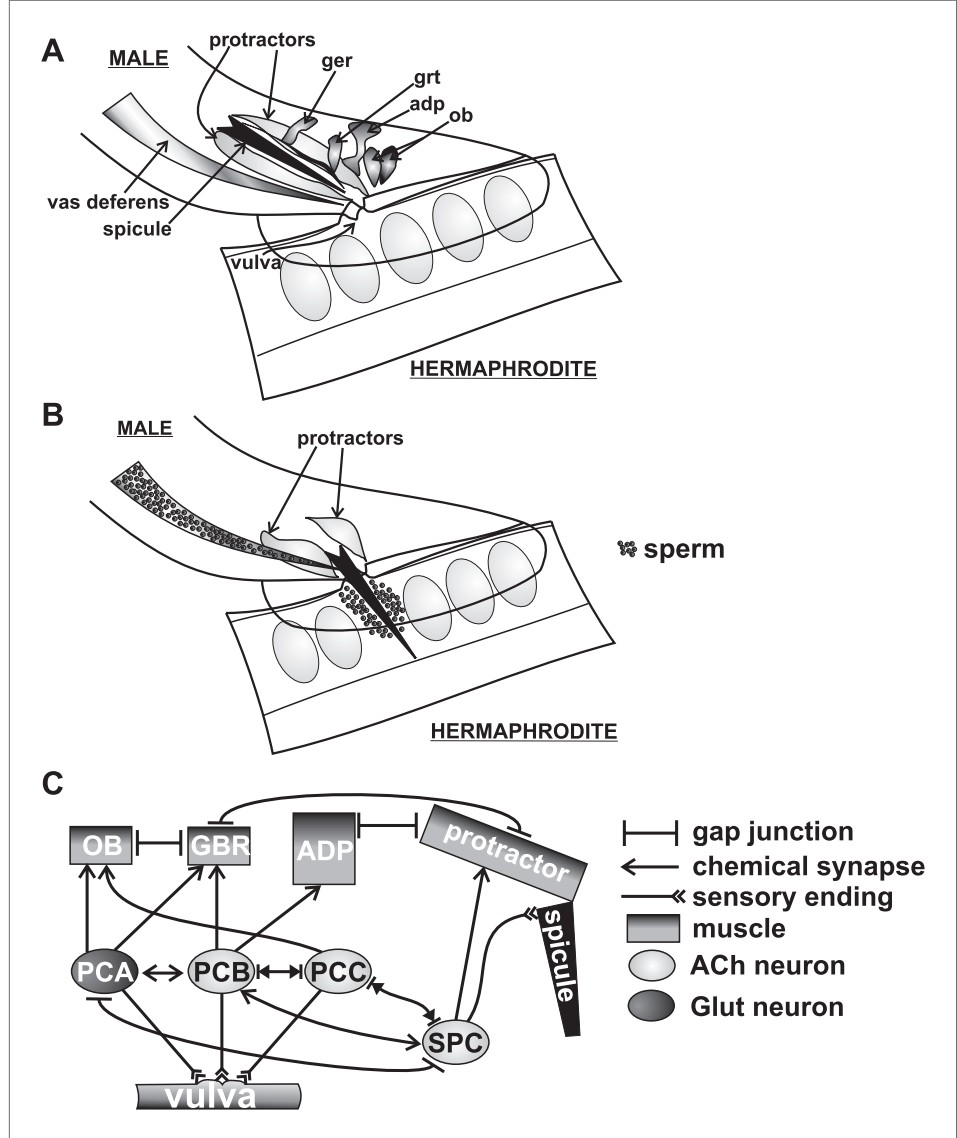

**Figure 1**. Conceptual diagram of structures and connectivity in the *C. elegans* male tail. ger = gubernaculum erector muscle, grt = gubernaculum retractor muscle, adp = anal depressor muscle, ob = oblique muscle. (**A**) Diagram of the male tail positioned at the hermaphrodite vulva. The positions of the copulatory spicules and associated muscles are indicated. The oval structures in the hermaphrodite depict eggs. (**B**) Diagram of the male during spicule insertion and sperm release. (**C**) Abridged connectivity in the male tail, adapted from *Jarrell et al. (2012)*.

The protractor contractions are transduced into spicule movements through their hemidesmosome attachments to the spicule cuticle (*Figure 1C*; *Sulston et al., 1980*; *Liu et al., 2011*). Appropriate prodding is limited to the vulva slit via dopamine (DA) signaling through the sensory ray neurons (*Correa et al., 2012*). The left-right bilateral cholinergic SPC proprioceptive neurons innervate the protractor muscles and have a sensory projection that is attached to the base of the spicules (*Figure 1C*). When the spicules partially penetrate the vulval slit, the SPC neurons induce the protractor muscles to contract tonically, likely through sensing the change in spicule position (*Garcia et al., 2001*; *Garcia and Sternberg, 2003*; *Liu et al., 2011*). ~14 s following spicule insertion, sperm moves from the seminal vesicle to the vas deferens (a process termed initiation) (*Figure 2A,B*), followed by drainage into the hermaphrodite uterus ~3 s later (a process termed release) (*Figure 1B*) (*Schindelman et al., 2006*).

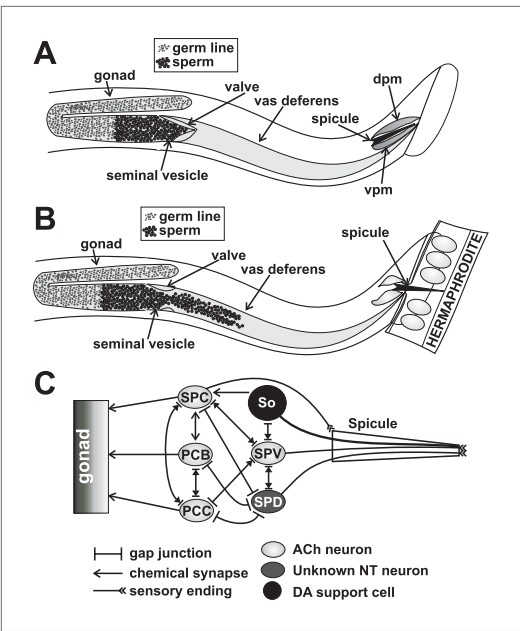

**Figure 2**. Conceptual diagrams of the structure and connectivity involved in ejaculation. (**A**) Diagram of the males' reproductive tract. dpm = dorsal protractor muscle, vpm = ventral protractor muscle. (**B**) Diagram of the initiation step of ejaculation. When the valve region separating the seminal vesicle from the vas deferens opens, sperm cells move toward the cloaca. (**C**) Connectivity of the spicule associated cells. Adapted from *Jarrell et al. (2012)*.

Since mating termination must occur during or following the ejaculation step, we hypothesize that cells which regulate ejaculation might contribute to how the males behave after coitus. Little is known of which cells initiate and control ejaculation. The SPC and p.c.s. neurons innervate the gonad, suggesting that they might regulate sperm movement (*Figure 2C*). Additionally, the SPV and SPD sensory neurons are proposed to inhibit premature ejaculation, but their modes of action remain unclear (*Liu and Sternberg, 1995*; *Schindelman et al., 2006*). These neurons send a sensory process down the two spicules' shafts and have sensory endings that are exposed to the environment at the spicule tips (*Figure 2C*; *Sulston et al., 1980*). They do not directly innervate the gonad, but they do have connections to the SPC and p.c.s., suggesting that they might indirectly regulate ejaculation through these neurons (*Figure 2C*; *Jarrell et al., 2012*).

In this study, we report that acetylcholine, dopamine, and glutamate secreting cells are stimulated upon successful intromission and promote sperm movement and release. Ejaculation, induced from the coordinated activities of the p.c.s., SPC, SPD, SPV, spicule socket cells, sex-muscles, and the gonad, leads to a refractory period consisting of reduced mating drive and ability. The duration of the refractory period allows the male time to recover, prevents him from mating multiple times with the same mate, and resets the complex neuromuscular network required for mating. Disruption of these cues leads to a shortened refractory period.

## Results

### Successful copulation is followed by a period of reduced mating drive and ability

Similar to other species, *C. elegans* males exhibit a period of reduced activity following ejaculation (*Barfield and Geyer, 1972*; *Oomura et al., 1983*; *Ureshi et al., 2002*). To develop metrics for measuring how the male's behavior changes immediately after mating, we first determined how fast *C. elegans* males re-copulate following ejaculation. Within 2 min after being placed on a 5 mm diameter bacterial lawn containing 15 immobile hermaphrodites, 1-day-old virgin males commenced mating and inserted their spicules into the hermaphrodites (*Figure 1B*, *Figure 3A*). However following ejaculation, ~12 min (SD = ±6 min 30 s) passed before they intromit their spicules again (*Figure 3A*). The males obviously regained the ability to mate prior to the 2nd intromission, but their individual mating behaviors were variable, preventing us from establishing a metric to determine effectively when their full ability to mate returned. Thus, we conservatively refer to the interval from the 1st spicule insertion to the 2nd spicule insertion as the 'refractory period'. We then asked if the refractory period duration was due to mating disinterest, reduced motor ability, or a combination of both.

To determine what behaviors were affected during the refractory period, we digitally recorded and measured the time males required to execute the different mating steps. Within 34 s after introduction to the mating lawn, virgin males placed their tail on the hermaphrodite cuticle and began searching for the vulva (*Figure 3B*); following intromission, they kept their spicules inserted for an average of 94 s. After spicule retraction, ~5 min passed before the males re-executed copulation (*Figure 3B*), indicating that male sexual drive was temporarily suppressed. However, even after males

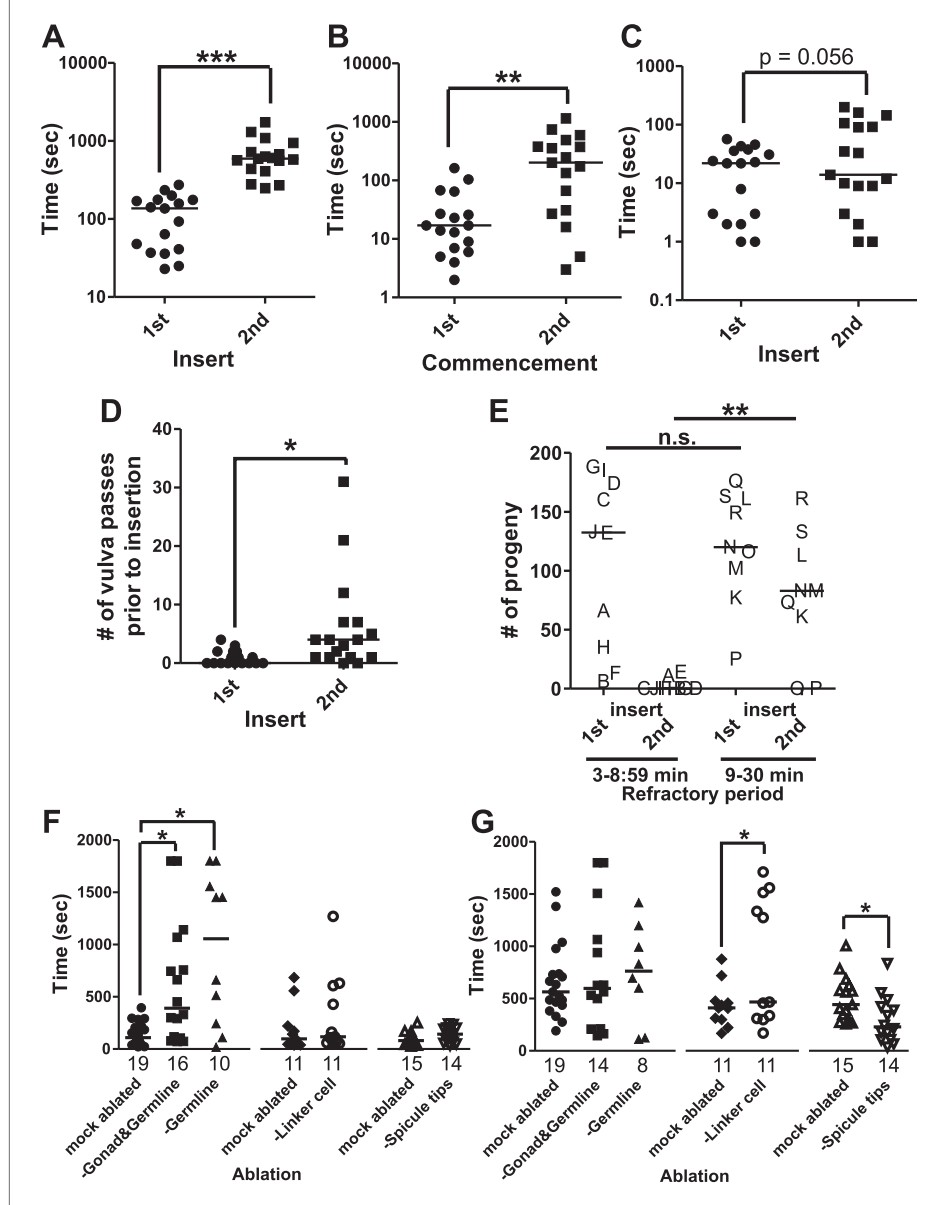

**Figure 3**. The refractory period is regulated by sperm release. Line represents median. (**A**) 1st insert: the time required for a male to insert his copulatory spicules into the hermaphrodite vulva from the time he was placed with the hermaphrodites. 2nd insert: the time from 1st insert to finding a second hermaphrodite and repeating the mating process (refractory period). y-axis, the time it takes a male to insert his spicules, x-axis, insertion number. (**B**) Mating drive. 1st commencement: the time it takes the male to begin mating with a hermaphrodite after being placed on a mating lawn. 2nd commencement: the time from retraction to the next beginning of mating. (**C**) The total time the male spent at the vulva prior to insert. (**D**) The number of vulva passes prior to insertion. (**A–D**) n = 17. *p<0.05, **p<0.005, ***p<0.0001, paired t test. (**E**) The number of progeny sired following successive couplings. Males are grouped into two categories: males that re-copulated between 3 and 8:59 min following the 1st insertion and males that re-copulated between 9 and 30 min following the first insertion. The letters identify the same male for each insertion. x-axis indicates the insert as well as what refractory period group each male was placed in. y-axis is the number of progeny each male sired for the indicated insert. **p<0.05, Mann–Whitney test. (**F** and **G**) The first insert (**F**) and refractory period (**G**) for males with the indicated cell(s) removed. *p<0.05, Mann–Whitney test. n is indicated below the x-axis. The x-axis indicates the cells removed during the operation, and the y-axis indicates the time it took for males to insert their spicules into the hermaphrodite.

recommended mating, the executions of subsequent motor steps were also slowed. The duration that the males spent positioned over the vulva prior to intromission was slightly longer, but not statistically different between the 1st and 2nd mating bouts (*Figure 3C*); however, the number of times they passed over the vulva was significantly increased (*Figure 3D*). These observations indicate that various male sensory-motor circuits recover differentially to the source(s) that attenuate mating behavior after ejaculation. This raised the question of why the male would need an extended time period between copulations.

In mammals, since the amount of sperm decreases in the male reproductive tract after ejaculation, the refractory period may provide an opportunity to re-establish sperm count (*Judd et al., 1997*; *Levin, 2009*; *Tlachi-Lopez et al., 2012*). To determine if, in *C. elegans*, a longer refractory period might increase the amount of sperm transferred, we counted the number of sired progeny as a proxy for successful ejaculation. Each male was allowed to mate two consecutive times, and the refractory period was recorded. Additionally, the cross-progeny from each mated hermaphrodite was counted to determine if the male had successfully ejaculated. We found that we could bin the males into two groups, the first of which took 3 to 8:59 min to copulate twice, and the second of which took 9 to 30 min to copulate twice. We found that males with a longer refractory period were more successful in siring progeny than their counterparts, whose refractory periods were shorter than 9 min (*Figure 3E*). Importantly, males from both groups did not display a difference in the amount of progeny they sired during the 1st mating (*Figure 3E*). This indicates that the reduced number of progeny sired during the 2nd mating, by the group of males with a shorter refractory period, is not due to a lack of ability. Thus, a longer refractory period allows males to recover their ability to sire progeny. We next asked how the various cellular components involved in ejaculation influence the refractory period.

## The sensing of sperm transfer is necessary for repeated mating success

We used the refractory period (the time between the 1st and 2nd spicule insertion) as a metric to identify the cellular source(s) that promotes post-coital mating attenuation. The male gonad was one of our candidates for regulating male sexual drive, since it is implicated in modifying whether males fed or searched for a mate (*Lipton et al., 2004*). The mature gonad consists of the germ line, seminal vesicle, and vas deferens (*Figure 2A*). We measured the initial insertion time (*Figure 3F*) and refractory period (*Figure 3G*) of males that had different cellular constituents of their gonad laser-ablated. We found that males lacking all of these structures required longer time to commence mating (*Figure 3F*), implying that the gonad sends a signal to promote this behavior. After the operated males inserted and retracted their spicules, their refractory period was similar to the mock-ablated control and coincidentally similar to the time required for their 1st spicule insertion (*Figure 3G*). The behavioral similarity between gonad-ablated males and post-coital intact males led us to speculate that a pro-mating gonadal signal might be repressed or depleted by an aspect of ejaculation; re-establishing the pro-mating signal might occur during the refractory period.

We next asked if the somatic gonad, consisting of the seminal vesicle (the storage area of mature sperm) and the vas deferens (a tubular seminal fluid-producing conduit for sperm movement), was sufficient to promote the initial mating drive, or if the germ line was also required. We found that germ line-ablated males behaviorally resembled the gonad-ablated males (*Figure 3F,G*), indicating that the germ line is required for the gonad to promote mating. Since the germ line produces sperm, we next asked if sperm production or sperm movement during ejaculation affects the initial mating drive and/ or the refractory period duration.

To uncouple sperm production from sperm release, we laser-ablated the linker cell in early larval stage 4 (L4, the final stage before adulthood) males. The linker cell guides the developing gonad to the posterior region of the male. Removal of the linker cell late in development prevents the mature vas deferens from connecting to the cloacal opening (*Sulston et al., 1980*). The somatic gonad and germ line are still functional and sperm moves from the seminal vesicle, but are retained in the vas deferens upon spicule insertion. Interestingly, the operation did not affect the initial mating drive (*Figure 3F*). However, as a population of operated males, preventing sperm release increased the duration of the refractory period, relative to the mock-ablated control (*Figure 3G*). Under closer inspection, the laser operation gave rise to two populations: males that displayed a normal refractory period and males that took much longer to re-mate (5 min 40 s (n = 6) vs 24 min 40 s (n = 5), p value=0.0043, Mann–Whitney test) (*Figure 3G*). In equal proportions, some males were able to

compensate for the lack of sperm movement, while others failed to do so. This suggests that the duration of the refractory period is plastic and could be modified by cellular components that directly or indirectly sense sperm release. This raised the question of what senses the transfer of sperm from the male into the hermaphrodite.

## SPV and SPD trigger sperm release and promote the refractory period

The male contains bilateral schlerotic copulatory spicules that he inserts into the hermaphrodite. The spicules serve to anchor the male to his mate and to widen the vulval channel to facilitate the flow of sperm. In addition, each spicule contains the sensory dendrites of two neurons, the bilateral left/right SPV and SPD (*Figure 2C*). These neurons send a process down the shaft of the spicule and have ciliated sensory endings that are exposed to the environment at the spicule tip. The processes of the neurons are surrounded by the structural sheath and socket cells (*Sulston et al., 1980*). When we laser-ablated these neurons during L4, the resulting males had variable mating problems. These defects include failure to follow through the steps of mating, failure to insert their spicules, and failure to ejaculate or prematurely ejaculating sperm prior to intromission (*Figure 4A*); some of these defects have been previously reported (*Liu and Sternberg, 1995*; *Schindelman et al., 2006*). These pleiotropic defects made measuring the refractory period difficult. We speculated that removing these neurons too early in larval development resulted in the variable re-wiring of the remaining circuits. To examine SPV and SPD's role during the refractory period, we used the laser to cut off the spicule tips in virgin adult males (*Figure 4—figure supplement 1*). This operation causes the cytoplasm of both SPV and SPD to leak out, killing the cells. Unlike the larval SPD and SPV-ablated males, spicule tip cut adult males did not display variable mating defects (*Figure 4A*). Both the mock-cut control and operated males displayed normal initial response times to hermaphrodites (*Figure 3F*), but interestingly, the spicule tips-cut males displayed a significantly shorter refractory period than control males ($296 \pm 217$ s vs $499 \pm 224$ s, p value=0.0121, Mann–Whitney test) (*Figure 3G*). This suggests that in wild-type males, the spicule sensilla modify the duration of the refractory period.

Although spicule tips-cut males lacking functional SPV and SPD could insert their spicules, many of the males were defective in ejaculation (*Figure 4A*). Analysis of sperm movement revealed defects in the initiation and release steps of ejaculation. Under standard conditions, sperm moves from the seminal vesicle through the valve to the vas deferens (initiation) (*Figure 2B*), and then drains out the cloaca into the hermaphrodite uterus (release) (*Figure 1B*; *Schindelman et al., 2006*). In approximately half of the spicule tips-cut males, the valve region opened (12/13 control vs 8/15 ablated, p=0.0377, Fisher's exact test), but sperm rarely passed out of the cloaca into the hermaphrodite (10/14 control vs 1/15 ablated, p=0.0005, Fisher's exact test). To verify if sperm transferred into the hermaphrodites, we moved the mated partner to individual plates after each successful insertion. The mated hermaphrodites contained a mutation that disrupts locomotion. We counted the number of plates that had moving cross-progeny, as a proxy for successful sperm release. 67% of control males sired progeny at the 1st mating and 33% of males sired progeny at the second mating (n = 12) (*Figure 4B*). No operated males were successful at the 1st insert, and only two were successful the 2nd time (n = 12) (*Figure 4B*; *Liu and Sternberg, 1995*; *Schindelman et al., 2006*). Thus, the putative sensory function of the spicules promotes both sperm movement and affects the refractory period. This raised the possibility that these two phenomena are coupled. The refractory state follows the ejaculatory behavioral state. Normally, the males should maintain the ejaculatory mating state, keeping their spicules inserted until all sperm are transferred (*Schindelman et al., 2006*). Therefore, we used spicule insertion time as a proxy metric to ask if the spicule sensilla regulate the duration of the ejaculatory mating state.

To determine if the spicule sensilla influence insertion duration, we recorded the 1st and 2nd copulations of virgin mock-cut and spicule tips-cut males. We did not measure any differences in mating drive or ability between the two populations (*Figure 4—figure supplement 2*). The average spicule insertion duration during the 1st mating was also not significantly different between the two treatments, although we observed a larger variability for spicule tips-cut males (F value = 0.0009) (*Figure 4C*). However, we noticed this difference became exaggerated during the 2nd mating. Generally, a male maintains spicule insertion throughout the sperm transfer process, which normally takes longer than 30 s. In contrast, many of the mating-experienced cut males inserted and retracted their spicules multiple times, with each penetration lasting only a few seconds (*Figure 4D*, *Figure 4—figure supplement 2*). This observation indicated that the spicule tips-cut males were defective in either entering or

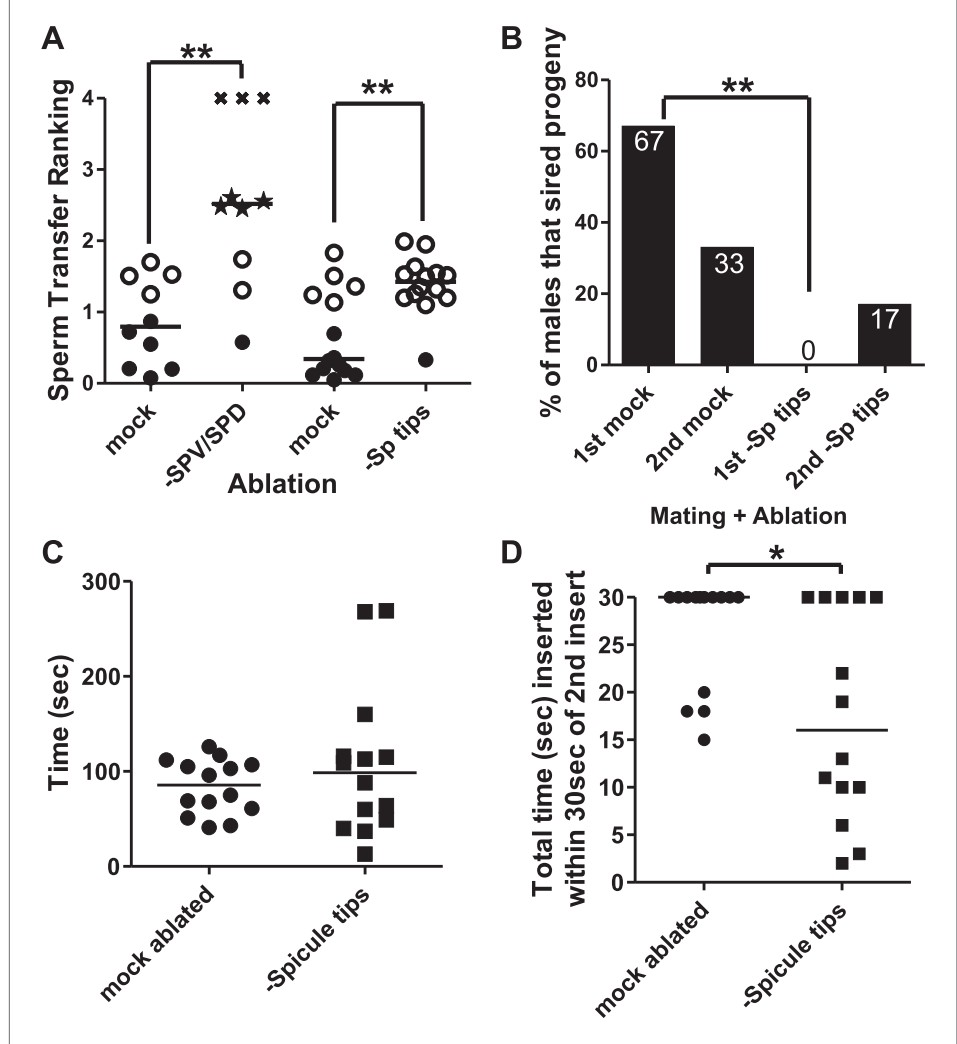

**Figure 4**. The spicule sensilla promote insertion behaviors. (**A**) Sperm transfer ranking for males lacking the SPV and SPD sensory neurons and their age-matched controls. A male received a 0 if he ejaculated into the uterus, a 1 if he inserted but didn't ejaculate, a 2 if he ejaculated without inserting into the vulva, and a 3 if he was unable to ejaculate or insert. Additionally, each male was given up to 5 min to insert and transfer sperm. That time was divided by 300 s and added to the number he received for ejaculating, giving him a final ranking. Filled circles represent males that inserted and transferred sperm into the hermaphrodite. Open circles represent males that inserted but did not transfer sperm. Stars represent males that ectopically ejaculated. Xs represent males that neither inserted nor ejaculated within 5 min. Line indicates mean. **p<0.005, Mann–Whitney test. (**B**) Mating potency. The number on each column is the % of males that sired progeny. **p<0.005, Fisher's exact test. *n* = 12. (**C**) Time the males remained inserted the first time they mated. (**D**) Following the second intromission, spicule tips cut males often did not leave their spicules inserted very long. Reported here is the total amount of time the males' spicules were inserted into the hermaphrodite for the first 30 s following the 2nd insert. *p<0.05, Mann–Whitney test. (**C** and **D**) *n* = 14.

The following figure supplements are available for figure 4:

**Figure supplement 1**. Cells affected by cutting off the spicule tips with a laser.

**Figure supplement 2**. Spicule tips cut males display defects in mating.

maintaining the ejaculatory state. Therefore, we were interested in identifying what additional cells or circuits function temporally with the spicule sensilla to transition the male between intromission, ejaculation, and refractory behaviors.

## SPC initiates sperm movement and the SPD and SPV facilitate sperm release

We utilized males that co-expressed the $Ca^{2+}$ sensor G-CaMP and the mDsRed fluorescent proteins to determine if cells, previously implicated through connectome wiring and laser-ablation/behavioral analyses, function during spicule penetration, sperm transfer, and post ejaculation behaviors (*Nakai et al., 2001*; *Tian et al., 2009*; *Correa et al., 2012*). As the males were mating, we digitally recorded the G-CaMP fluorescence changes and later measured and corrected them using the mDsRed fluorescence as a non-fluctuating reference. The data were plotted as a $\%\Delta F/F0$ over time and correlated with the male's behavior. A representative male is shown for each cell(s) of interest. Figure supplements contain four additional males that inserted their spicules and one male that prod at the vulva but did not insert.

To measure $Ca^{2+}$ transients in the SPV and SPD neurons during mating, we expressed G-CaMP and mDsRed in these cells using the *gpa-1* promoter (P*gpa-1*) (*Jiang and Sternberg, 1999a*). In many males, individual SPV and SPD fluorescence could not be isolated, since their cell bodies overlapped. Thus we measured their combined fluorescence during data analysis. After spicule insertion, $Ca^{2+}$ transients in these cells gradually increased; the average peak activity occurred ~4 s post intromission. The $Ca^{2+}$ transients began to decline during the sperm's movement through the vas deferens and continued to decrease after sperm release (n = 5, *Figure 5A*; *Table 1*; *Figure 5—figure supplement 1*). Although the data indicate that the SPV and SPD are active post intromission and during the initial stages of ejaculation, we surveyed additional cells to see if their activities are also correlated with the timing of spicule insertion and ejaculation.

Previous work found that sustained spicule insertion requires the bilateral cholinergic SPC sensorimotor neurons. From their structural morphology, the neurons are speculated to sense spicule movements through their proprioceptive attachments to the base of the spicules. Via acetylcholine (ACh)-mediated synaptic transmission, the SPC neurons can then induce tonic spicule protractor muscle contraction (*Figure 1C*). Laser-ablation of these cells impairs sustained spicule intromission and consequently, the ability to ejaculate. The SPC makes chemical synapses to the gonad, raising the possibility that these neurons might also contribute to sperm movement (*Figure 2C*; *Sulston et al., 1980*; *Liu and Sternberg, 1995*; *Garcia et al., 2001*; *Jarrell et al., 2012*). We expressed G-CaMP in the SPC using the P*gar-3B* promoter and recorded fluorescent changes during mating (*Liu et al., 2007*). Coincident with full spicule penetration, $Ca^{2+}$ transients rapidly increased in the SPC. The peak G-CaMP fluorescence intensity occurred faster than in the SPV and SPD neurons (1.3 s vs 4.0 s, p<0.05, ANOVA, *Figure 4B*; *Table 1*; *Figure 4—figure supplement 1*). However, similar to the SPV and SPD, the $Ca^{2+}$ transients in the SPC began to decrease during sperm movement into the gonadal vas deferens (*Figure 5B*).

The somatic gonad stores sperm until receiving appropriate cues (*Figure 2A,B*). Upon spicule insertion, this organ facilitates sperm transfer to the hermaphrodite (*Figure 1B*). The gonad's active role in ejaculation is not well understood, although it requires the cellular secretion machinery to promote sperm release (*Schindelman et al., 2006*). Video recordings of the sperm-activating protease TRY-5 highlight the fluid movement in the gonad. First, fluid containing TRY-5 within the vas deferens transfers to the hermaphrodite, and then additional TRY-5 release coincides with sperm movement from the valve (*Smith and Stanfield, 2011*). Since the cholinergic SPC makes chemical synapses to the somatic gonad, we asked if ACh signaling can stimulate gonadal $Ca^{2+}$ transients. The potentially non-specific ACh agonist oxotremorine S (OxoS) induces males to protrude their spicules and some males to ejaculate. We expressed G-CaMP in the somatic gonadal valve using the *try-5* promoter (*Smith and Stanfield, 2011*) and asked if exogenously applied OxoS can induce valve $Ca^{2+}$ transients. We saw increases in gonadal $Ca^{2+}$ transients in males (n = 5) that were exposed to the agonist (*Figure 5C*), indicating that the somatic gonad activity can be stimulated by ACh. We then asked when during mating can we detect somatic gonad activity.

Prior to spicule insertion, the gonadal valve keeps the sperm in the seminal vesicle. Immediately upon spicule insertion, we measured that the valve's $Ca^{2+}$ transients reached their peak (~1.8 s) with kinetics similar to the SPC neurons (*Figure 5D*; *Table 1*; *Figure 5—figure supplement 1*). This coordination in cellular activities is consistent with the idea that signaling from the cholinergic SPC partially contributes to the initiation of ejaculation (*Garcia et al., 2001*; *Emmons, 2005*). However, although $Ca^{2+}$ transients in the valve increased upon intromission, the contractions that open the valve did not

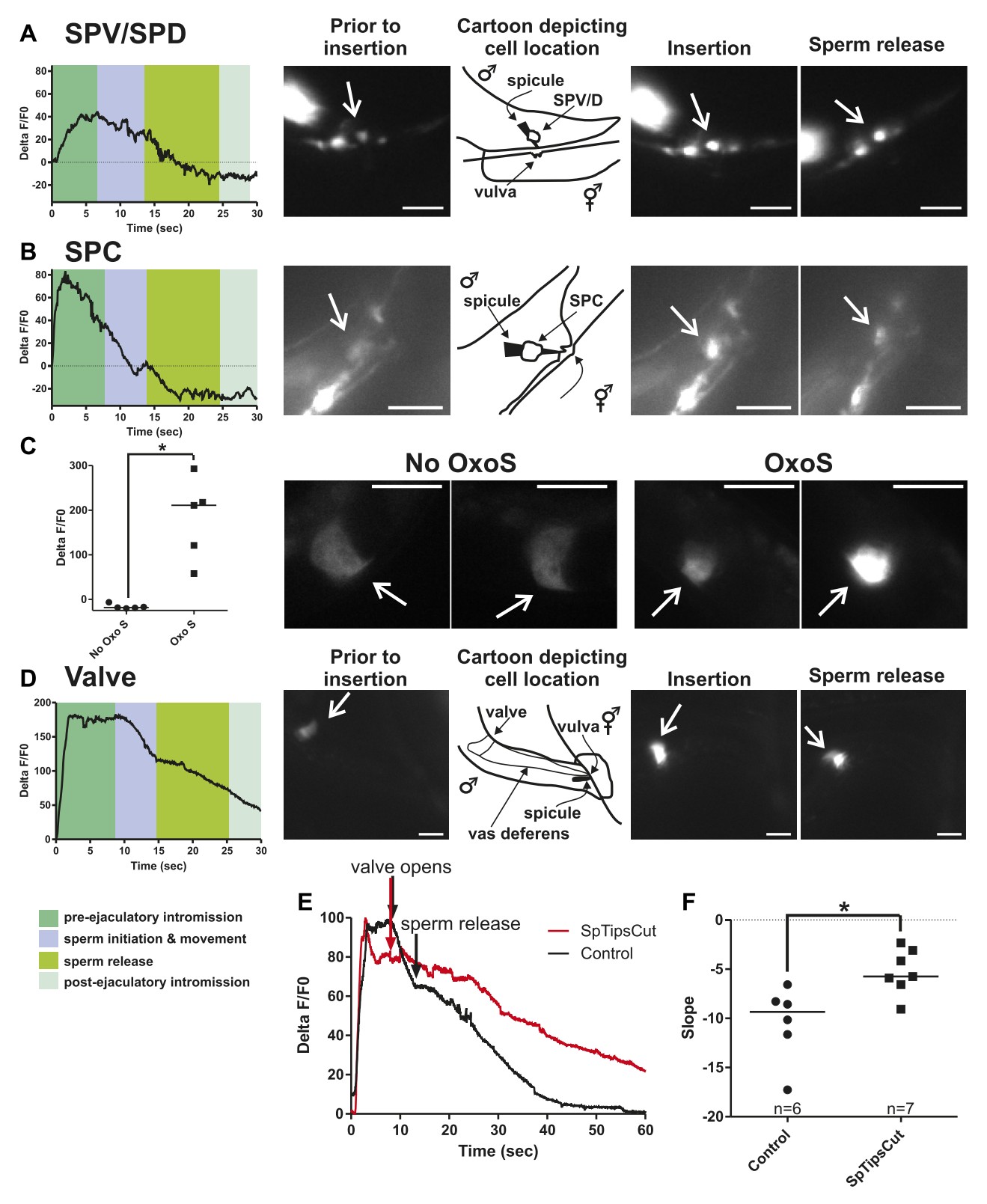

**Figure 5**. Ca²⁺ transients in SPV, SPD, SPC neurons, and gonad muscles during intromission and ejaculation. (**A**, **B**, **D**). % ΔF/F0 trace including insertion and ejaculation. One representative recording is reported for each cell type. x-axis is time (seconds), y-axis is % ΔF/F0. The time of mating behaviors is indicated in color. Additional traces given in *Figure 4—figure supplement 1* and Figure 4—figure supplement 3. (**A**) Trace of the SPV/SPD neurons, *Figure 5. Continued on next page*

*Figure 5. Continued*

G-CaMP expressed from P*gpa-1*. Scale bar = 20 µM. (**B**) Trace of the SPC neuron, G-CaMP expressed from P*gar-3B*. Scale bar = 10 µM. (**C**) Graph of the largest calcium change ($\Delta$F/F0) from time point 0. The pictures display fluorescence in the valve region at time 0 and 1 min later for males exposed and not exposed to OxoS. Males express G-CaMP in the gonadal valve from P*try-5*. *p value<0.05, Mann–Whitney test. Scale bar = 10 µM. (**D** and **E**) Traces of the valve region of the gonad, G-CaMP expressed from P*try-5*. Scale bar = 100 µM. SpTipsCut = spicule tips cut. (**F**) Slope of the line extrapolated from the highest $Ca^{2+}$ transient plus 15 s. This time was chosen to include sperm release. *p value<0.05, Mann–Whitney test. Line represents median.

The following figure supplements are available for figure 5:

**Figure supplement 1**. $Ca^{2+}$ transient changes in cells that regulate ejaculation.

**Figure supplement 2**. Removing the spicule sensilla impacts the $Ca^{2+}$ changes in the gonadal valve.

**Table 1.** $Ca^{2+}$ transients following spicule insertion

| Cell | ↑$Ca^{2+}$ insertion→peak (sec)* | Significance | Slope of initial $Ca^{2+}$ increase ($\Delta$F/F0%/sec) | Significance |
|---|---|---|---|---|
| SPC | 1.3 ± 0.52[a] | | 45 ± 13 | |
| Valve | 1.8 ± 0.79[a] | | 60 ± 14 | |
| SPV/SPD | 4.0 ± 1.0 | p<0.05 to 'a' | 21 ± 14 | p<0.05 to SPC and valve |
| PCA | 6.3 ± 0.56 | p<0.05 to 'a' | 15 ± 2.9 | p<0.05 to SPC and valve |
| Socket cells | 1.7 ± 0.36[a] | | 94 ± 18 | p<0.005 to PCA |
| Sex muscles 1st peak | 1.3 ± 1.7[a] | | 155 ± 82 | p<0.005 to socket cells |
| Sex muscles 2nd peak | 11 ± 3.9 | p<0.05 to sex muscles 1st peak | 46 ± 19† | p<0.0001 to sex muscles 1st peak |
| SPC control | 1.2 ± 0.30 | | 60 ± 26 | |
| SPC ablated | 2.2 ± 0.74 | p=0.022 to SPC control | 29 ± 12 | p=0.035 to SPC control |

Mean and standard deviation reported. For non-operated cell types, *n* = 5. Results of ANOVA: Newman–Keuls Multiple Comparison Test are shown. SPC–control and ablated, Mann–Whitney *t* test. SPC–control *n* = 6, SPC–ablated *n* = 7.
*Time (sec) required for $Ca^{2+}$ to increase from spicule insertion to $Ca^{2+}$ peak.
†Slope determined from where the $Ca^{2+}$ begins to increase for a second time to $Ca^{2+}$ peak following this second increase.
[a]Peak times that are significantly different from the SPV/SPD and PCA cells.

immediately occur. $Ca^{2+}$ transients remained high in the valve until it opened, on average 6.9 ± 1.5 s after insertion (n = 7). When sperm moved through the vas deferens, the level of $Ca^{2+}$ transients in the valve began to slowly decrease (*Figure 5D*). The delay between spicule insertion and the movement of sperm out of the seminal vesicle suggests that additional cells, which are active during this interval, might contribute to the opening of the valve.

Since the $Ca^{2+}$ transient levels in the SPV and SPD neurons peaked after the SPC neurons, we asked if damaging these sensory cells, by using a laser to cut off the spicule tips, would interfere with the opening of the valve or perturb the profile of the valve's $Ca^{2+}$ transients. Similar to the mock-cut males, the valve in spicule-tips cut males displayed a rapid increase in $Ca^{2+}$ transients (*Figure 5E*, *Figure 5—figure supplement 2*). But instead of a distinct decrease in $Ca^{2+}$ transients that coincide with the opening of the valve and sperm movement, the decrease in $Ca^{2+}$ transients was more gradual and protracted (*Figure 5E,F*). Coincident with the altered activity profiles, the valve did not open in 10/19 males (compared to 18/19 controls, p value=0.0078, Fisher's exact test), and in the remaining males, sperm was not transferred to the hermaphrodite. In contrast, nearly all mock-cut controls released sperm into their mates (1/19 ablated vs 18/19 controls, p value<0.0001, Fisher's exact test). Taken together, the $Ca^{2+}$ imaging suggests that immediately upon intromission, the SPC primes the gonad

for the initiation of ejaculation, and the spicule sensory neurons provide further stimulation necessary for sperm movement and release.

## Glutamatergic PCA signals sex muscle contractions resulting in sperm release

The calcium imaging and laser-ablation experiments suggested that during mating, the SPV, SPD, and SPC neurons facilitated ejaculation. Therefore, we wanted to confirm if stimulating a limited number of cells that included these neurons is sufficient to induce ejaculation in non-mating males. To address this question, we utilized the short wavelength light (blue, 475 nm)-activated ion channel channel-rhodopsin2 (ChR2) (*Nagel et al., 2003, 2005*). We expressed ChR2 from the promoters: P*gar-3b* (expresses in multiple cells including the SPC) (*Liu et al., 2007*), P*unc-103*B (many cells including the post cloacal sensilla and spicule sensilla and sex muscles) (*Reiner et al., 2006*), P*unc-17* (all cholinergic neurons) (*Alfonso et al., 1993*; *Garcia et al., 2001*), and P*unc-17*S (head cholinergic neurons only). We found that none of the transgenic males expressing ChR2 initiated ejaculation or released sperm upon

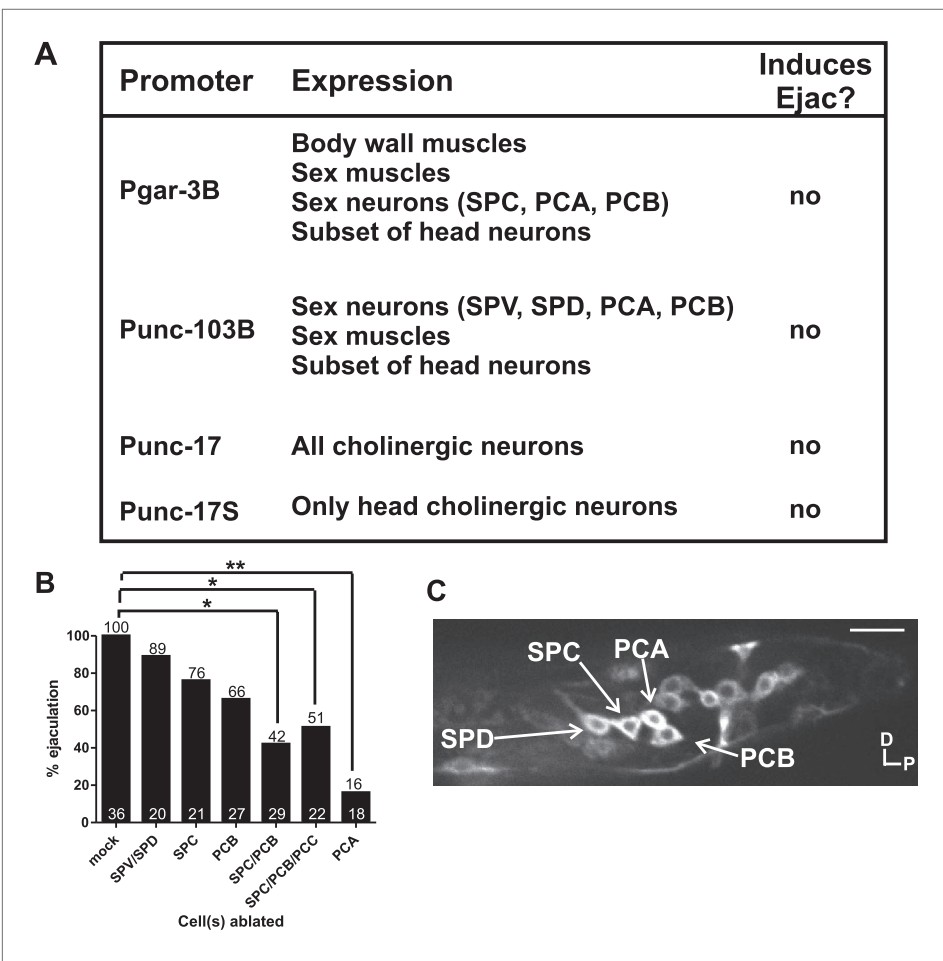

**Figure 6**. Cholinergic and glutamatergic neurons promote ejaculation. (**A**) Chart of promoter, expression pattern, and ability of the promoter driving ChR2 to induce ectopic ejaculation (Ejac). (**B**) Graph of ectopic ejaculation in response to 475 nm wavelength light in rgIs6[*Punc-17 small:ChR2, Punc-103B:ChR2, Pgar-3B:ChR2*] males. *x*-axis indicates the cells ablated, *y*-axis is the percent of males that ectopically ejaculated in response to 475 nm wavelength light stimulation. *n* numbers are indicated at the bottom of each bar. *$p<0.05$, **$p<0.005$, Fisher's exact test. (**C**) Fluorescent, confocal image of a rgIs6[*Punc-17 small:ChR2, Punc-103B:ChR2, Pgar-3B:ChR2*] L4 male tail. Dorsal is up, posterior to the right. Cells important in mating behavior are indicated by the arrows. Scale bar = 10 µM.

The following figure supplements are available for figure 6:

**Figure supplement 1**. % of ChR2-expressing males that ejaculate in response to 475 wavelength light stimulation.

stimulation (*Figure 6A*). This result indicated that stimulation of specific groups of cells is not sufficient to induce the ejaculatory state. However, transgenic males simultaneously expressing P*unc-103*B:ChR2, P*unc-17*S:ChR2, and P*gar-3*B:ChR2 did ejaculate upon light stimulation (*Figure 6B*). This indicated that a larger number of cells than we expected, which includes the SPD, SPC, PCA, and PCB neurons, as well as sex-shared head and ventral cord neurons and sex muscles, must be stimulated to initiate ejaculation and release sperm (*Figure 6C*).

Evoking ejaculation in non-mating males via intense 475 nm wavelength light and ChR2 is artificial, but we used the assay to help us identify additional cells that might be involved in ejaculation behavior. We laser-ablated various ChR2-expressing cells to determine if artificial ejaculation could be perturbed. We killed the SPV/SPD and SPC neurons singly and found that these treatments slightly reduced the ejaculation efficiency, but not enough to be statistically significant (p value=0.78 and 0.42, respectively, Fisher's exact test) (*Figure 6B*, *Figure 6—figure supplement 1*). Since removing the spicule sensilla and SPC did not significantly reduce artificial ejaculation, we next asked if other mating-associated cells in the male tail, specifically the post cloacal sensilla, can contribute to the behavior.

The bilateral PCA, PCB, and PCC neurons constitute the post cloacal sensilla. They redundantly function to sense the vulva and execute repetitive intromission attempts (*Liu and Sternberg, 1995*; *Liu et al., 2011*). The cholinergic PCB and PCC make chemical synapses to the gonad (*Figure 2C*; *Garcia et al., 2001*; *Jarrell et al., 2012*). We initially focused on the PCB, since it expresses ChR2 and PCC does not. We found that ablating the PCB singly reduced the ejaculation efficiency, but like the SPC neurons, not enough to be statistically significant (p value=0.2, Fisher's exact test) (*Figure 6B*). However, compared to the mock-ablated control, ablating both SPC and PCB significantly reduced the number of males that released sperm (42%, p=0.013, Fisher's exact test) (*Figure 6B*). This suggested an additive role for these cells in ejaculation behavior. However, some males still ejaculated in response to 475 nm light stimulation. This indicates that ejaculation-promoting neurons remain; this could include the post cloacal sensilla PCC and PCA.

In the transgenic males, the PCC neurons did not express ChR2, but nonetheless, we laser-ablated the PCC with SPC and PCB. We reasoned that PCC could be indirectly activated by light-induced ChR2 stimulation through their connectivity with other cells in the circuit (*Figure 2C*; *Jarrell et al., 2012*). However, we did not see any obvious decrease in ejaculation behavior when comparing SPC/PCB/PCC to SPC/PCB ablated males (p value=0.37, Fisher's exact test) (*Figure 6B*). This raised the possibility that the remaining post cloacal sensilla, the PCA neurons, could be involved in promoting ejaculation.

Unlike the SPC, PCB, and PCC, the PCA neurons do not make chemical synapses with the gonad and are not cholinergic. However, similar to the PCB and PCC, PCA does make motor synapses to the posterior sex muscles (*Figure 1C*; *Jarrell et al., 2012*). To our surprise, we observed that few PCA-ablated males were able to ejaculate after stimulation (16%, p value=0.0003, Fisher's exact test) (*Figure 6B*). This uncovered a role for the PCA in promoting ejaculation.

The PCA, PCB, and PCC neurons make chemical synapses to the male specific oblique and gubernaculum muscles (*Figure 1C*). The oblique muscles, when contracted, help the male press the posterior region of his tail over the vulva during intromission attempts. During ejaculation, the contracted gubernaculum muscles pull the male proctodeum/protracted spicules posteriorly to allow the vas deferens' opening access to the cloacal opening (*Figure 7A*). Laser-ablation of the gubernaculum muscles causes the protracted spicules to block the vas deferens, thus inhibiting sperm release (*Liu et al., 2011*). We speculated that if the PCA neurons facilitate sperm release, then the gubernaculum muscles and the PCA might display correlated activity. We used the P*unc-103E* promoter to express G-CaMP in the male sex muscles, and focused our region-of-interest on the anal depressor, gubernaculum erector, and gubernaculum retractor (*Reiner et al., 2006*). As expected, the sex muscles displayed a peak in Ca$^{2+}$ transients immediately upon spicule intromission (*Table 1*; *Figure 7A,B*, *Figure 7—figure supplement 1*). However, the Ca$^{2+}$ transients slightly decreased after the initial spicule penetration. Coincident with the initiation of ejaculation, ~11 s later, the gubernaculum and anal depressor muscles displayed an increase in Ca$^{2+}$ transients and became hyper-contracted (*Table 1*). After ~15 to 20 s, we detected sperm in the hermaphrodite uterus. When sperm transfer was completed, the Ca$^{2+}$ transients in the muscles began to oscillate as they dampened (*Figure 7B*).

We used the P*eat-4* promoter to express G-CaMP in the PCA neurons to determine if the activity of these cells coincided with ejaculatory gubernaculum muscle contractions (*Figure 7—figure supplement 1*). The PCA is likely glutamatergic, since it expresses the *eat-4* encoded glutamate vesicular

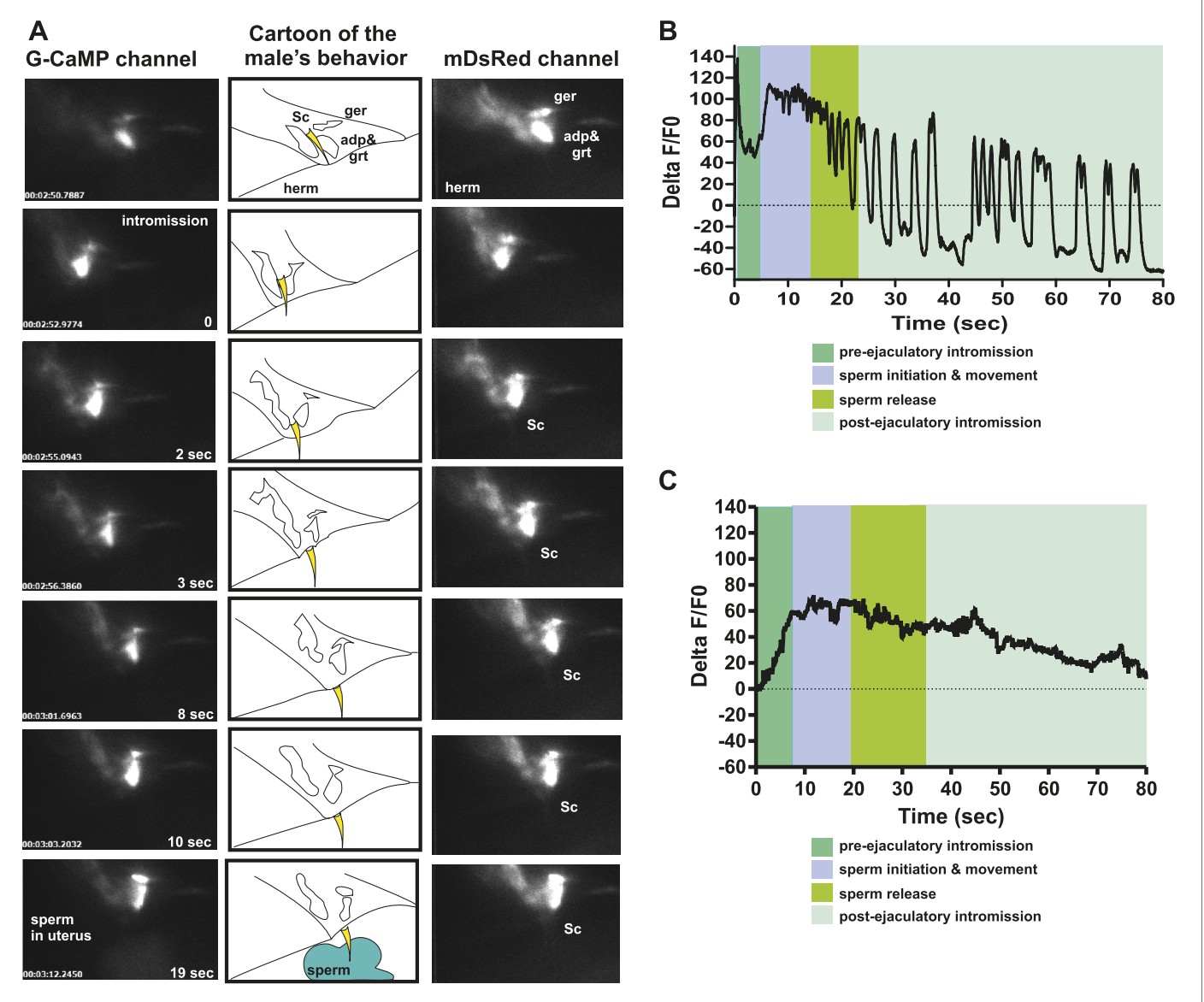

**Figure 7**. PCA contributes to the sex muscle-controlled spicule movement required for sperm to drain from the cloaca into the uterus. (**A**) Images of sex muscle activity during intromission and ejaculation. Sc = spicule, ger = gubernaculum erector, adp = anal depressor, grt = gubernaculum retractor. The gubernaculum is located posteriorly to the spicule and assists in spicule movement. After insertion and prior to sperm release, the gubernaculum erector is required to adjust spicule position to allow sperm to drain into the hermaphrodite. The image on the left indicates G-CaMP fluorescence, the central image of diagram of male tail position at the hermaphrodite vulva, and the right image is mDsRed fluorescence. Time is indicated on the left image and spicule location on the right. (**B**) % ΔF/F0 in the anal depressor and the gubernaculum erector and retractor. G-CaMP expressed from P*unc-103E*. (**C**) % ΔF/F0 in the post cloacal sensilla PCA. G-CaMP expressed from P*eat-4*.

The following figure supplements are available for figure 7:

**Figure supplement 1**. Ca²⁺ transient changes in sex muscles and the PCA neurons during ejaculation.

transporter (*Lee et al., 1999*, *2008*; *Serrano-Saiz et al., 2013*). We found that the PCA Ca²⁺ transients gradually increased after spicule insertion with their first peak at 6.3 ± 0.56 s post intromission (n = 5, *Table 1*; *Figure 7C*, *Figure 7—figure supplement 1*); but unlike the SPC and the SPV/SPD, elevated Ca²⁺ transients lasted after sperm release (*Figure 7C*). The timing of the peak PCA Ca²⁺ transients (9.2 ± 2.9 s) proceeded and overlapped with the second Ca²⁺ increase in the gubernaculum erector and retractor and anal depressor muscles (11 ± 3.9 s, p value=0.8, Mann–Whitney t test compared to

PCA peak), suggesting PCA is promoting these changes. However, laser-ablation of the PCA did not have gross effects on vulva location behavior, spicule intromission, or ejaculation. Likely, the PCB and PCC neurons can compensate for the damaged cells in operated males (*Liu and Sternberg, 1995*; *Liu et al., 2011*).

## SPC neuronal activity promoted by SPV and SPD initiates sperm release

The ChR2 experiments revealed a role for the PCA, SPC, and PCB neurons in regulating ejaculation. Surprisingly, ablating the SPV/SPD neurons in this experiment did not have an effect, suggesting ChR2 is expressed in downstream targets of SPV/SPD. These neurons share chemical and electrical synapses with SPC and electrical synapses with PCB, and could indirectly regulate PCA (*Figure 2C*; *Jarrell et al., 2012*). To test this hypothesis, we cut off the spicule tips in males expressing G-CaMP in the PCA, sex muscles, SPC, and PCB and asked if removing the SPV/SPD neurons impacted Ca$^{2+}$ transients during mating.

We utilized P*eat-4* and P*unc-103E* to drive G-CaMP expression in the PCA neurons and the dorsal sex muscles, respectively. Removing the SPV/SPD neurons did not affect PCA or dorsal sex muscle activity (*Figure 8A,B*, *Figure 8—figure supplements 1, 2*). Thus, PCA and its targets are not downstream of SPV/SPD. To determine what could be a downstream target, we analyzed the SPC motor neuron, which shares chemical and electrical synapses with SPV and electrical synapses with SPD (*Figure 2C*; *Jarrell et al., 2012*).

We utilized P*gar-3B*:G-CaMP-expressing males to examine the effect of removing SPV/SPD on Ca$^{2+}$ transients in the SPC neurons. Upon insertion, the Ca$^{2+}$ transients in the SPC increased rapidly, followed by a decline to near-baseline levels around the time of sperm release (*Table 1*; *Figure 8C,D*, *Figure 8—figure supplement 3*). In contrast, when SPV/SPD were removed, the increases occurred at a slower rate (*Table 1*; *Figure 8C,D*, *Figure 8—figure supplement 3*). This suggests that the SPV/SPD are promoting SPC activity upon insertion. However, the ChR2 experiments revealed that removing the SPC alone is insufficient to reduce 475 nm wavelength light-induced ejaculation, suggesting that SPV/SPD may have more than one target.

Removing the PCB neurons in conjunction with the SPC neurons was able to reduce 475 nm wavelength light-induced ejaculation in ChR2-expressing males. Thus, we analyzed how removing the SPV/SPD impacted Ca$^{2+}$ transients in the PCB. We utilized P*dop-2* to drive G-CaMP expression in the PCB (*Correa et al., 2012*). Ca$^{2+}$ transients in the PCB displayed minor increases upon insertion, and their activity quickly dissipated (*Figure 8—figure supplement 4*). This activity was unaffected by SPV/SPD removal (*Figure 8E*, *Figure 8—figure supplement 4*). This result suggests that PCB does not play a significant role after insertion. However, it does synapse to the gonad and can respond to SPD activity through its electrical coupling. We propose that in SPC-ablated males, the PCB can compensate for the loss of the SPC and promote 475 nm wavelength light-induced ejaculation in ChR2-expressing males.

## Dopamine released from the spicule support cells promotes sperm release and the refractory period

The severing of the spicule tips, which removes the SPV and SPD neurons in adult *C. elegans* males, also affects the neuronal support sheath and socket cells (*Figure 4—figure supplement 1*, *Figure 9—figure supplement 1*). In adult males, the socket cells encase the neuronal processes and sheath cells inside the spicule (*Sulston et al., 1980*). Four socket cells are located on each side of the male tail, and during L4 larval development they secrete the sclerotic material that hardens to make the spicules (*Jiang and Sternberg, 1999b*). We asked if removing these socket cells negatively impacted the male's ability to mate.

We laser-ablated all eight spicule socket cell nuclei in adult males to determine how this operation impacted mating behavior. We found that insertion was unaffected (11 out of 27 control mock-ablated males could insert, compared to 7/22 ablated males, p value=0.6, Fisher's exact test). However, only 1/7 ablated males released sperm into the hermaphrodite for ≥10 s; the other ablated males showed variable dysfunctions in executing or sustaining sperm release. In contrast, 73% (n = 8/11) of mock-ablated males released sperm into the hermaphrodite for ≥10 s (p value=0.0498, Fisher's exact test). Thus, similar to cutting off the spicule tips, ejaculation was significantly reduced, indicating that the socket cells promote sperm release. However, ablating the socket cells was not equivalent to cutting off the spicule tips. In socket cell ablations, 6 out of 7 (86%) males released sperm from the valve (vs 12/12, 100% for mock-ablated), compare to only 8 out of 15 (53%) spicule tips-cut males (vs 12/13,

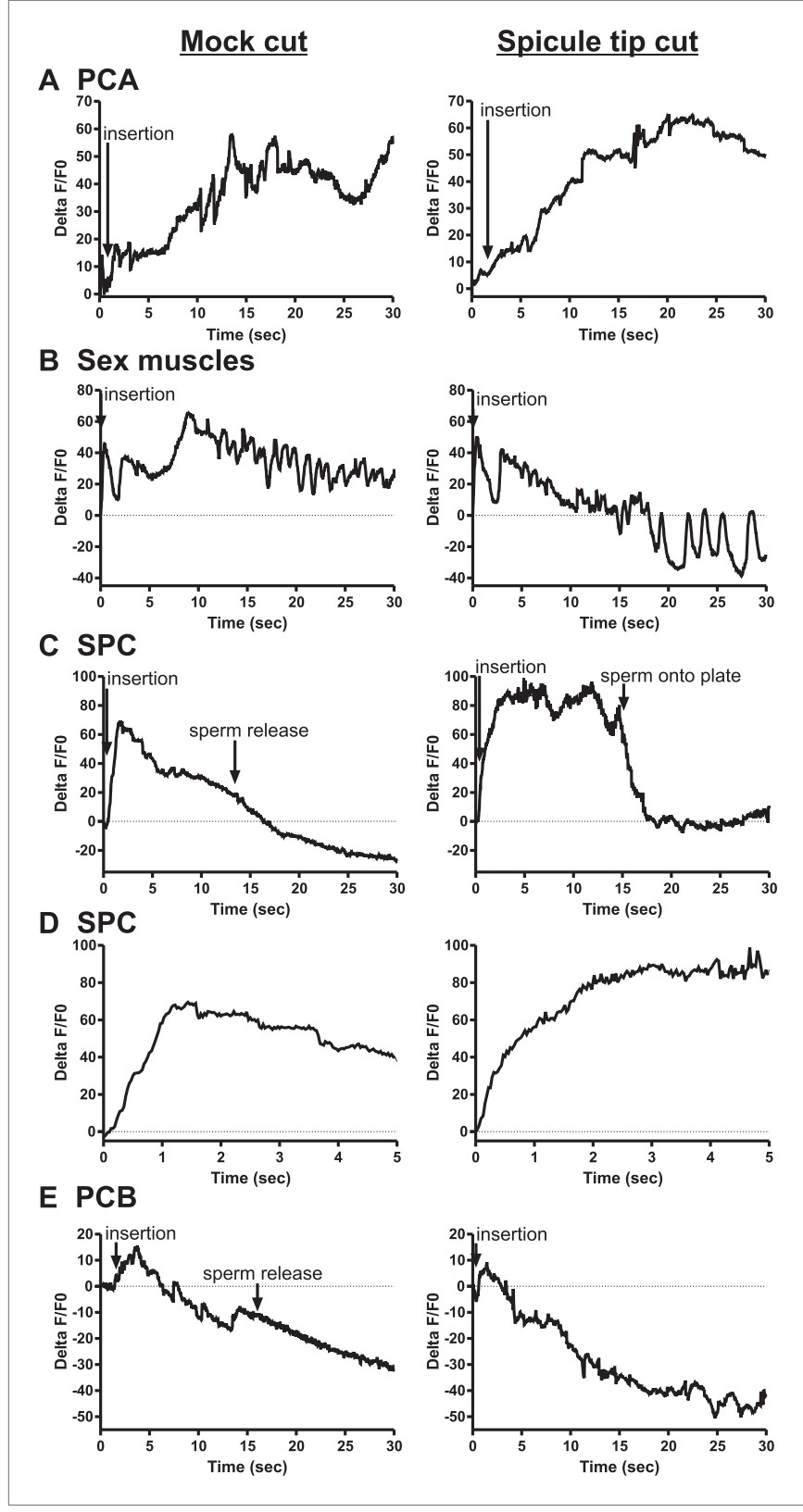

**Figure 8**. Calcium imaging in spicule tip cut males. (**A**) % ΔF/F0 in the PCA. (**B**) % ΔF/F0 in the dorsal protractor, anal depressor, and gubernaculum erector and retractor. (**C** and **D**) % ΔF/F0 in the SPC. (**E**) % ΔF/F0 in the PCB.
*Figure 8. Continued on next page*

*Figure 8. Continued*

The following figure supplements are available for figure 8:

**Figure supplement 1**. Removing the spicule sensilla has no effect on the PCA neuron Ca$^{2+}$ transients.

**Figure supplement 2**. Removing the spicule sensilla has no effect on the sex muscle Ca$^{2+}$ transients.

**Figure supplement 3**. Ca$^{2+}$ transients increase at a slower rate in the SPC neurons when the spicule sensilla are removed.

**Figure supplement 4**. Removing the spicule sensilla has no effect on the PCB neuron Ca$^{2+}$ transients.

92% mock cut control, p value=0.038, Fisher's exact test). Thus, removing the SPV and SPD in conjunction with the socket cells had a more significant impact on mating behavior.

We favored the idea that the spicule socket cells directly function in ejaculation behaviors, but we also considered the hypothesis that perturbing these structural cells might indirectly affect the function of neurons or other cells that regulate sperm release. A prediction from this hypothesis is that during intromission and ejaculation behavior, these cells are inert and will display no obvious changes in their cellular activities. To explore these ideas, we expressed G-CaMP in the socket cells using the P*bas-1* promoter and observed the socket cells during mating behavior. *bas-1* encodes the aromatic amino acid decarboxylase that converts L-DOPA to the neurotransmitter dopamine (DA) in neurons (*Hare and Loer, 2004*). We found that socket cell Ca$^{2+}$ transients increased rapidly upon insertion, with a profile similar to the SPC and the gonadal valve (*Table 1*; *Figure 9A*, *Figure 9—figure supplement 2*). Ca$^{2+}$ transients then declined to about half their peak intensity at the time of sperm release (*Figure 9A*). This supports the hypothesis that the socket cells are directly involved in ejaculatory behaviors. We next asked through what mechanism might the neuronal support cells be promoting sperm release.

Intriguingly, the enzyme necessary to make DA, tyrosine hydroxylase (*cat-2*), in addition to aromatic amino acid decarboxylase (*bas-1*), is also expressed in these neuronal support cells (*Figure 9—figure supplement 1*; *Lints and Emmons, 1999*; *Hare and Loer, 2004*). Thus, we analyzed the contribution of the neurotransmitter DA in socket cell function during mating behavior. Previous work identified the importance of DA in the sex-specific bilateral sensory ray neurons R5A, R7A, and R9A in modulating tail posture, backward locomotion, and intromission attempts (*Sulston et al., 1975*; *Koo et al., 2011*; *Correa et al., 2012*). Additionally, DA deficient mutants exhibit difficulty transferring sperm, but the DA source was not identified (*Correa et al., 2012*). To determine if the DA-expressing rays might potentially promote sperm movement in addition to the socket cells, we expressed G-CaMP using the P*dat-1* promoter. We found that in contrast to the socket cells, these neurons exhibited no gross changes in Ca$^{2+}$ transients after spicule insertion (*Figure 9B*, *Figure 9—figure supplement 2*). This suggests that if DA promotes ejaculation, its source would be the socket cells, not the ray neurons.

To determine if DA regulates ejaculation, we looked at the behavior of tyrosine hydroxylase deficient *cat-2(lf)* mutant males. We previously reported that these mutants execute ectopic intromission attempts on non-vulval areas of the hermaphrodite and have difficulty inserting their spicules into the vulva (*Correa et al., 2012*). Additional work reported a role for DA in switching between behavioral states (*Sawin et al., 2000*; *Vidal-Gadea et al., 2011*). In this study, we observed that the few *cat-2(lf)* males capable of spicule insertion displayed additional defects. Many mutant males had difficulty releasing sperm or sustaining sperm release into the hermaphrodite; some males even displayed 'coitus interruptus', by pulling out their spicules from the hermaphrodite and then erratically releasing sperm onto the media. When we measured if the males can sustain sperm release into the hermaphrodite for at least 10 s, 79% of wild-type males successfully transferred sperm, compared to only 31% of *cat-2(lf)* males (p=0.02, Fisher's exact test, *Figure 9C*).

To verify that the mating defects were due to the lack of tyrosine hydroxylase and not because of some undescribed developmental defect, we asked if treating mutant adults with exogenous DA can ameliorate the *cat-2(lf)* mating defects. We exposed *cat-2(lf)* adult males to exogenous DA for at least 3 hr and asked if mating ability was restored. The dopaminergic rays express the DA reuptake transporter (*Jayanthi et al., 1998*; *Koo et al., 2011*), but the socket cells do not, leaving open

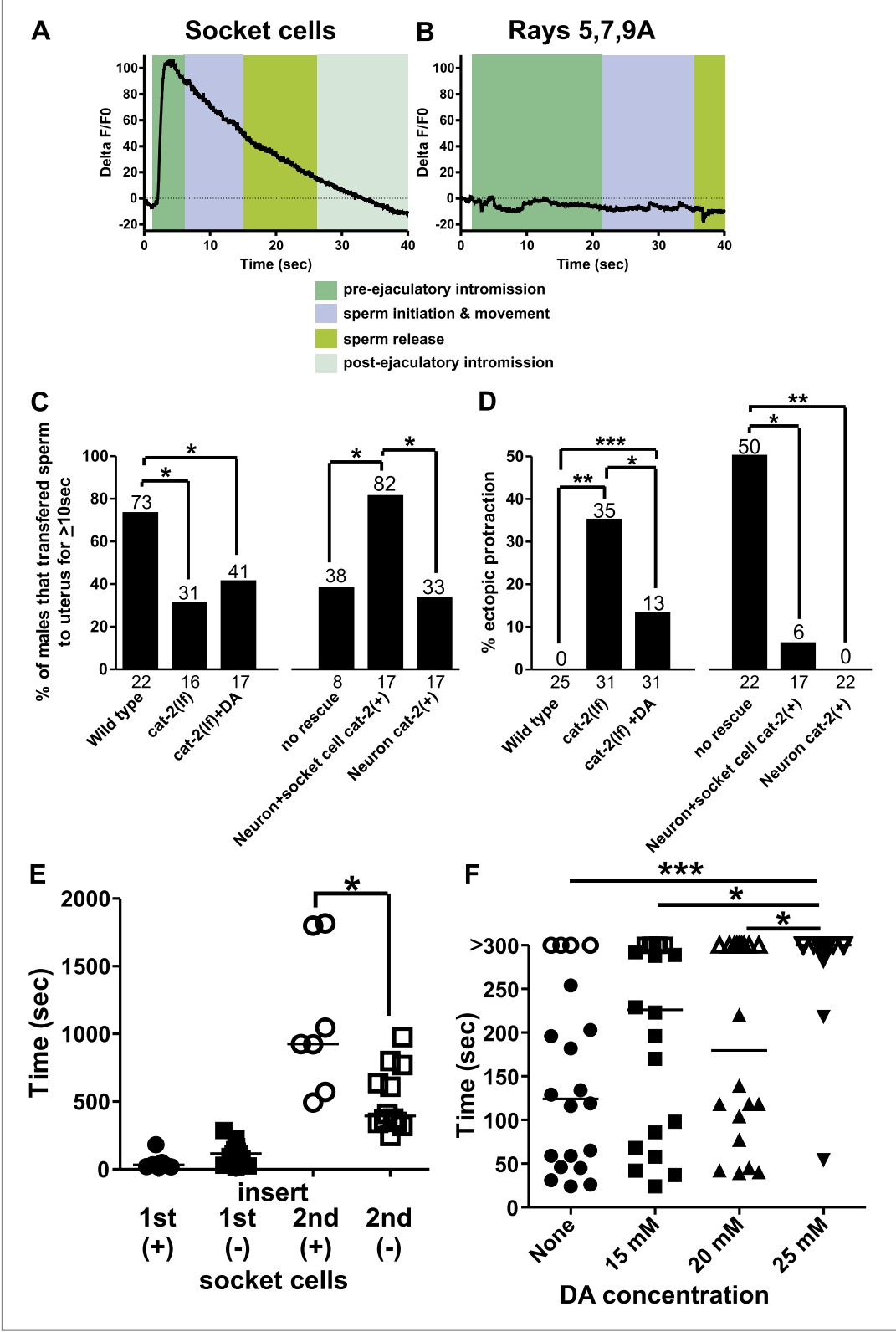

**Figure 9**. Socket cell dopamine (DA) promotes sperm release. (**A**) % ΔF/F0 in the socket cells. G-CaMP is expressed from P*cat-2*. (**B**) % ΔF/F0 in the dopaminergic rays 5,7,9A. G-CaMP is expressed from P*dat-1*. (**C**) % of males that transferred sperm for ≥10 s into the uterus. (**D**) Ectopic protraction exhibited when males are placed with
*Figure 9. Continued on next page*

*Figure 9. Continued*

hermaphrodites. (**C** and **D**) cat-2(+) indicates in what cells *cat-2* was rescued. *p value<0.05, **p value<0.005, ***p value<0.0001, Fisher's exact test. *n* values below the *x*-axis, percentages above the bars. (**E**) Time to 1st insert (1st) and refractory period (2nd) for *cat-2(lf)* rescued males. *cat-2(+)* is in all DA ray neurons. *p value<0.05, Mann–Whitney test. (**F**) The time required for virgin males to insert their spicules into a hermaphrodite. *x*-axis is the concentration of DA on which the males mated. *y*-axis is the amount of time (sec) the males took from being placed with hermaphrodites until they inserted their spicules into the uterus. Closed symbols indicate time from placement with hermaphrodites to insertion. Open symbols indicate the male did not insert within 5 min of being placed with hermaphrodites. Bars represent the median. *p value<0.05, **p value<0.005, ***p value<0.0001, one-way ANOVA with Bonferroni's correction.

The following figure supplements are available for figure 9:

**Figure supplement 1**. Socket cell DA regulates male mating.

**Figure supplement 2**. Ca$^{2+}$ transients in the DA-expressing cells in the male tail.

the possibility that exogenous DA might preferentially restore different aspects of mating performance. As expected, adult *cat-2(lf)* males exhibit reduced ectopic intromission attempts in response to exogenous DA (*Figure 9D*). This result is consistent with our previous study suggesting that the dopaminergic ray neurons regulate the spicule protraction circuit. However interestingly, exogenously applied DA did not restore the ability to transfer sperm (*Figure 9C*).

To determine which dopaminergic cells promoted sperm release, we expressed *cat-2*(+) in the dopaminergic rays and socket cells using the P*cat-2* promoter; this construct rescued both ectopic intromission attempts and ejaculation defects (*Figure 9C,D*). We next used the P*dat-1* promoter to express *cat-2* in ray neurons, but not in the spicule socket cells. Similar to exogenous DA exposure, ectopic intromission attempts were suppressed to wild-type levels (*Figure 9D*), but sperm transfer remained abnormal (*Figure 9C*). Thus, DA synthesis in the socket cells is necessary for sperm release in *C. elegans* males. Since our previous experiments established a link between sperm release and the refractory period, we then asked if the socket cell DA could also regulate the post-ejaculatory refractory period.

The *cat-2(lf)* male's intromission efficiency is severely compromised by his erratic ectopic intromission attempts. This made it difficult for us to obtain enough males to compare their refractory periods (the interval between the 1st spicule insertion and the 2nd spicule insertion) to wild-type males with statistical confidence. However, P*dat-1* expression of *cat-2(+)* in the ray neurons restored the male's intromission ability, and allowed us to score the spicule socket cell's contribution to the refractory period (*Figure 9—figure supplement 1*). We found that during the first mating, transgenic *cat-2(lf)* males expressing *cat-2(+)* in the ray neurons and spicule socket cells, or just in the ray neurons commenced mating in a similar time frame (51 s vs 121 s, p value=0.08, Mann–Whitney test) (*Figure 9E*). However, transgenic males that lacked *cat-2*(+) in the socket cells had shorter refractory periods, relative to males with *cat-2*(+)-containing socket cells (518 s vs 1083 s, p value=0.0127, Mann–Whitney test) (*Figure 9E*). Thus, DA contributes to the refractory period length. This result is consistent with our previous observations that the refractory period and the efficiency of sperm release correlate with the functions of the spicule-associated cells.

Since endogenous DA extends the period of time between copulations, we asked if exogenous DA is able to extend the time required for a virgin male to mate successfully the 1st time. Exogenous DA is capable of paralyzing hermaphrodites after 20 min of exposure on an NGM plate supplemented with 25–30 mM DA (*Chase et al., 2004*). To avoid paralyzing the males, we gave them 5 min to mate on plates supplemented with various concentrations of DA. We did not know which concentration would extend a virgin male's first mating experience and therefore tested a variety of concentrations. We found that while concentrations of 15 and 20 mM DA slowed down the time it takes a male to insert his spicules, the difference was not statistically significant (*Figure 9F*). However, 25 mM DA induces a significant increase in the time required for males to insert their spicules (*Figure 9F*). Indeed, few males were successful in inserting their spicules within this time frame. This decreased mating ability was not due to sluggish movement or paralysis, as they still commenced mating in a similar, but slightly longer timeframe (*Figure 9—figure supplement 1*) and their backing velocity along the hermaphrodite remained the same (171 ± 43 μm/s *n* = 6 control, 178 ± 35 μm/s *n* = 5 25 mM DA). Thus, exogenous DA is capable of interfering with males' mating ability.

## Discussion

Much remains to be discovered concerning the molecular and structural pathways involved in post ejaculatory behavioral activity (*Levin, 2009*; *Turley and Rowland, 2013*). Mating is energy-costly and precludes participation in other behaviors such as feeding (*Schneider et al., 2013*). Thus, a period of reduced activity following successful copulation is advantageous to organisms. In rats, the post ejaculatory interval represents the period (~6–10 min) between ejaculations. After at least five successive ejaculations, males achieve a state of sexual satiation. During satiation, male rats will not mate again for 6–14 days (*Beach and Jordan, 1956*; *Phillips-Farfan and Fernandez-Guasti, 2009*). In this work, we also measured the post ejaculatory period in *C. elegans* males, referred to as the refractory period. However, we do not know if *C. elegans* males achieve a state of sexual satiation as exhibited by rodents. *C. elegans* males display an average lifespan of 12 days, show significant decline in sexual function at day 3, and are unable to sire progeny by day 5 (*Gems and Riddle, 1996*; *Guo et al., 2012*). Therefore, the advantages *C. elegans* males would have by exhibiting an extended satiation time after multiple intromissions are uncertain, since their functional reproductive span is very short.

We discovered that a longer refractory period allows males to recover their ability to transfer sperm. When males attempt to mate again too quickly, their ability to sire progeny is reduced. Although males recover from the refractory period whether they can sire progeny or not, we favor the hypothesis that the refractory period ensures that a male does not continuously copulate with the same mate. Following ejaculation, the males' mechanosensory and chemosensory neurons are still in immediate proximity to the hermaphrodite cues. A male that reinitiates the mating program, only to intromit his spicules into the same partner, is at a competitive disadvantage to disseminate his genetic material. Thus, the period of reduced activity and ability we observed following ejaculation may help insure that the male copulates with different partners (*Figure 10A*).

We determined that the refractory period is dependent upon the males' ability to produce and sense successful ejaculation. These interconnected activities partly depend on the SPV and SPD neurons. These neurons' sensory endings are exposed to the intrauterine environment through the spicules tips (*Figure 2C*; *Sulston et al., 1980*), allowing them to respond after insertion and during sperm release. When we cut off the spicule tips (*Figure 4—figure supplement 1*), removing the neuronal and support cell function, males displayed a reduced ability to transfer sperm and recovered from mating quicker. We propose that SPV and SPD sense the intrauterine environment, promote sperm release, and regulate the refractory period.

The tight coupling of ejaculation and the refractory period led us to examine how sperm initiation and release are controlled in the male (*Figure 10A*). Calcium imaging data support a role for the SPV and SPD in regulating sperm release. However, these cells do not synapse the gonad (*Figure 2C*). Rare males lacking these cells are able to initiate sperm movement from the valve, but they poorly control the release of sperm into the hermaphrodite, and generally spill their ejaculate onto the mating lawn. Therefore, the SPV and SPD contribute to both sperm initiation and release (*Figure 10B*), providing control of the timing of sperm movement (*Liu and Sternberg, 1995*).

Our data indicate that the SPC neurons, with support from the SPV/SPD neurons and the post cloacal sensilla, trigger the initiation of ejaculation (*Figure 10B*). The initial ejaculation step occurs when the valve opens, permitting sperm to move from the seminal vesicle to the vas deferens (*Figure 2B*; *Schindelman et al., 2006*). The cholinergic SPC was previously identified as the trigger for full spicule intromission, once the hermaphrodite vulva is breached. However, since ablating the SPC results in males that cannot insert their spicules, evidence for their role in ejaculation was lacking (*Liu and Sternberg, 1995*; *Garcia et al., 2001*). Our calcium imaging and optogenetic data reveal SPC's role as a trigger of sperm initiation. The SPC displays a rapid increase in $Ca^{2+}$ transients immediately upon insertion. Additionally, ablating SPC in conjunction with the post cloacal sensilla neuron PCB results in ChR2-expressing males that do not ejaculate in response to light activation. We propose a model where the SPC, via direct cholinergic signaling to the gonad (*Jarrell et al., 2012*), triggers the initiation of ejaculation (*Figure 10B*). This interaction allows coupling of full spicule insertion into the uterus with the initiation of sperm movement.

Interestingly, we found that the rapid increase in SPC $Ca^{2+}$ transients upon insertion is dependent on the SPV and SPD neurons, despite the fact that the SPV and SPD do not reach their peak activity as fast as the SPC. The SPC shares chemical synapses with the SPV and electrical synapses with the SPV and SPD (*Figure 2C*; *Jarrell et al., 2012*). We propose that the SPV and SPD help maintain a lower

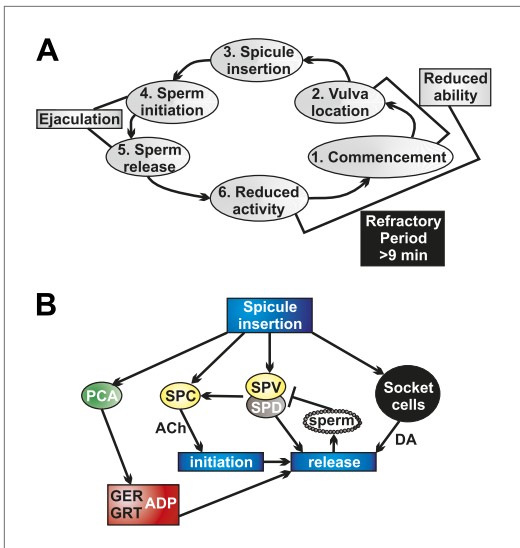

**Figure 10**. Socket cell DA and neuronal ACh regulate ejaculation and the refractory period. (**A**) The steps of *C. elegans* male mating behavior. The order of the individual steps is given by the numbers. The refractory period is a period of reduced activity and mating ability following ejaculation. (**B**) Neurons, muscles, and support cells that are activated by spicule insertion. Mating behavior steps are represented by blue boxes. GER = gubernaculum erector muscle, GRT = gubernaculum retractor muscle, ADP = anal depressor muscle, DA = dopamine, ACh = acetylcholine. Glutamate is a possible PCA neurotransmitter.

threshold in the SPC, allowing the SPC to trigger spicule intromission without inducing premature ejaculation. When the spicules tips enter the uterine environment, the SPV and SPD neurons further facilitate the SPC activity to begin sperm initiation. Thus, without the SPV and SPD present, the SPC cannot reach its peak activity as quickly as normal. This model accounts for the role of SPV and SPD in SPC activity and initial sperm movement.

Once sperm initiation has occurred, and sperm has traveled the length of the vas deferens to the cloaca, it must be released into the hermaphrodite uterus (*Figure 1B*). Calcium imaging data presented in this work and ablation data in previous work suggest that the gubernaculum erector and retractor muscles, and possibly the anal depressor, are involved in the penultimate step of male mating (*Figure 10B*). The gubernaculum erector and retractor are located dorsally to the spicules and attach the gubernaculum to the body wall (*Figure 1A*). The gubernaculum is a thin strip of cuticle material located dorsally from the spicules and is thought to assist in positioning the copulatory structures (*Figure 1A*, *Figure 9—figure supplement 1*; *Sulston et al., 1980*). Our calcium imaging of the gubernacular and anal depressor muscles reveal that they undergo contraction and display increased $Ca^{2+}$ transients immediately prior to sperm transfer, suggesting this is a mechanism by which sperm is released. Previous work showed that males lacking the gubernaculum erector and retractor had difficulty transferring sperm, with sperm being stuck at the distal end of the vas deferens, unable to drain into the hermaphrodite (*Liu et al., 2011*). Together, these data indicate that the gubernaculum muscles contract to facilitate sperm release, likely by re-adjusting the spicules' position.

What promotes gubernacular muscle contraction and, ultimately, sperm release? Our data suggest the post cloacal sensilla PCA is partially responsible (*Figure 10B*). This neuron sends a process to the posterior of the cloacal opening, allowing it to sense and maintain vulva position, and also makes synapses with the gubernaculum erector and retractor (*Figure 1C*; *Liu and Sternberg, 1995*; *Liu et al., 2011*; *Jarrell et al., 2012*). We identified glutamate as a potential PCA neurotransmitter and showed that PCA's peak activity following intromission corresponds with gubernacular muscle contractions. Additionally, we showed that ChR2-expressing males were unable to ectopically ejaculate when PCA was removed. While ablating the PCA in a wild type mating background does not affect sperm transfer (*Liu and Sternberg, 1995*; *Liu et al., 2011*), we propose this is due to the ability of the other p.c.s. neurons, the PCB and PCC, which make similar muscular connections (*Jarrell et al., 2012*), to compensate for the loss of the PCA. Both redundancy and the ability to compensate when one function has been impaired is reported in the male mating circuit in general and the p.c.s. neurons specifically (*Liu and Sternberg, 1995*; *Koo et al., 2011*; *Liu et al., 2011*; *LeBoeuf and Garcia, 2012*). Additionally, p.c.s. function likely contributes to the ability of exceptional males lacking the SPV/SPD/socket cells to initiate sperm release. The synaptic connections they make to the gonad could function to prime this organ when the p.c.s. are active during prodding behavior, increasing the likelihood that impaired SPC function could nevertheless stimulate sperm initiation. Thus, the sex muscle contractions facilitated by the PCA and other neurons are a necessary component of sperm release.

However, PCA activity and gubernaculum sex muscle contractions are insufficient for sperm release into the uterus. Males that display normal activity in these cells still fail to transfer sperm in the absence

of the SPV/SPD and socket cells. Our data indicate that the SPV/SPD neurons are involved in sperm initiation and possibly release, while the DA-expressing socket cells play no role in initiation but are required for sperm release (*Figure 10B*). Removal of the socket cells themselves or their ability to produce DA results in impaired sperm release. These data support an evolutionarily conserved role for DA in promoting ejaculation (*Peeters and Giuliano, 2008*). This role of DA could represent a general theme of monoaminergic modulation of sexual motor acts, as there is also peripheral modulation of the corpus cavernosum in mammals by monoamines, including DA and serotonin (*Angulo et al., 2001*; *Hayes and Adaikan, 2002*, *d'Emmanuele di Villa Bianca et al., 2005*; *Senbel, 2011*).

Interestingly, the socket cells represent a non-neuronal source of DA. The socket cells in *C. elegans* are proposed to be invertebrate glia (*Ward et al., 1975*; *Oikonomou and Shaham, 2011*). Since impairing glia function is known to negatively impact *C. elegans* neuronal function (*Bacaj et al., 2008*; *Felton and Johnson, 2011*), we cannot rule out that disrupting socket cell function impairs the SPV/SPD. To our knowledge, a role for neuronal support cells in either DA production or ejaculatory behavior has not been previously identified. Additionally, socket cell DA may function hormonally, as these cells do not express the DA re-uptake transporter. Vesicles have been identified in socket cells in the hermaphrodite head, suggesting this cell type is secretory (*Doroquez et al., 2014*). Extrasynaptic roles for DA, as important modulatory components of neurotransmission, have been identified in both *C. elegans* and other organisms, and DA receptors appear on tissues throughout animals (*Sargent et al., 1977*; *Chase et al., 2004*; *Asano et al., 2012*; *Fuxe et al., 2012*). Our results suggest a functional role for extraneuronal and extrasynaptic DA in coordinating ejaculation. Additionally, our experiments show that DA promotes the refractory period, the time between successful mating attempts, coupling sperm release with mating-related behavioral inhibition.

We propose a model where the DA that promotes ejaculation, either humorally or synaptically, acts to reduce the activity of the post-copulatory male. Once the effect of DA is slowly extinguished, the male is then able to mate again. The refractory period could indirectly cause males to mate with different hermaphrodites. However, the refractory period could also provide the gonad and male mating circuitry time to recover, as we observed that mating drive returns before mating potency (*Figure 10A*).

This raises the question of what does the male mating circuitry need to recover from? One possibility is that different circuits involved in mating need varying times to re-set following the activities necessary for successful ejaculation. Strikingly, we noticed that a large number of neurons and other cells show a dramatic increase in calcium transients upon spicule insertion. This suggests that a large amount of neurotransmitters, neuropeptides, and hormones are released at intromission. While the large amount of information contained within these signals is required for sperm release, over a period of time they may have a dampening effect on the circuitry. Many receptors undergo periods of desensitization, and the large amounts of signals released near-simultaneously might compound this effect. The re-setting of circuits could require the removal of signaling molecules that inundate the male's nervous system during copulation. Thus, the receptors could return to functional levels required for subsequent mating. Additionally, calcium transients oscillate in the sex muscles following the initiation of sperm movement. While we do not know the purpose for these fluctuations, they could be in response to the signals released upon intromission, and their dampening is likely required for re-insertion. Thus, the high amount of circuit activity required for ejaculation might necessitate a similar period of reduced activity to restore proper circuit function.

We propose that the SPV and SPD neurons not only promote sperm initiation but regulate the length of the refractory period. When these neurons' sensory endings enter the uterus, their activity increases and consequently sperm initiation occurs. However, their activity decreases when they sense sperm release (*Figure 10B*). Consequently, the activity of the ejaculation circuit, which includes the SPC and socket cells, is also reduced. If the SPV and SPD do not sense intrauterine sperm, then their extended activity subsequently promotes the continued release of DA and other neurotransmitters. This could result in the lengthening of the refractory period. This hypothesis is based on the interesting observation we obtained when we prevented sperm from draining into the hermaphrodite but did not interfere with SPV and SPD function. This experiment resulted in a subpopulation of males that displayed an extended refractory period. Thus in intact wild-type males, if the SPV and SPD do not efficiently sense sperm in the uterine environment, a possible cause could be due to a low ejaculate sperm count. Consequently, lengthening the refractory period would allow time for the males to produce sperm.

In mammals, post-ejaculatory regulation of the refractory period occurs in networks in the hypothalamus, amygdala, and septal nuclei (*Gogate et al., 1995*; *Parfitt and Newman, 1998*; *Dominguez and*

*Hull, 2010*). In contrast, we identified that the refractory period in *C. elegans* is modulated by the circuitry located in the male tail. However, we cannot rule out a role for the head ganglia in regulating both ejaculation and the refractory period. We were unable to achieve optogenetically controlled ejaculation unless ChR2 was expressed in cholinergic head neurons. This suggests that, similar to other species, males' sexual activities are modulated by neuronal networks that are not directly associated with the sex organs.

Additional parallels can be drawn between the circuits regulating ejaculation in mammals and the circuits we have discovered in *C. elegans*. The dorsal nerve in rats and humans that is part of the copulatory organ elicits ejaculatory responses (*Pescatori et al., 1993*; *Wieder et al., 2000*). This signal is sent to lumbar spinothalamic (LSt) neurons in the spinal cord (*Truitt and Coolen, 2002*). LSt cells then signal the neurons that control the emission and expulsion phases of ejaculation. The emission phase involves the secretion and movement of seminal fluids to the proximal urethra via seminal vesicle and vas deferens contraction. Once the ejaculate is in position, it is ejected from the penis in the expulsion phase that includes the rhythmic contractions of smooth muscle (*Coolen, 2005*). *C. elegans* ejaculatory circuitry is similarly set up (*Figure 10*) and allows us to expand the general understanding of circuit control of ejaculatory and post-ejaculatory behaviors.

## Materials and methods

### Strains

Animals were maintained on NGM agar plates with *E. coli* strain OP50 and contain *him-5(e1490)* for their high instance of males (*Brenner, 1974*; *Hodgkin et al., 1979*). Strains used in this study were: *let-23(sy1)* (*Aroian and Sternberg, 1991*) on LGII, *pha-1(e2123)* (*Schnabel and Schnabel, 1990*) and *unc-64(e426)* (*Brenner, 1974*) on LGIII, rgIs6[*Punc-17 small:ChR2::YFP, Punc-103B:ChR2::YFP, Pgar-3B:ChR2::YFP*] on LGIV, and *lite-1(ce314)* on LGX (*Edwards et al., 2008*).

Transgenic strains include: rgEx494[*Punc-17 small:ChR2::YFP, Punc-103B:ChR2::YFP*], rgEx501[*Punc-17 small:ChR2::YFP, Pgar-3B:ChR2::YFP*], rgEx485[*Punc-17 small:ChR2::YFP*], rgEx496[*Punc-17 small: ChR2::YFP, Punc-103B:ChR2::YFP, Pgar-3B:ChR2::YFP* TL1], rgEx506[*Punc-17 small:ChR2::YFP, Punc-103B:ChR2::YFP, Pgar-3B:ChR2::YFP* TL2], rgEx551[*Punc-17 small:ChR2::YFP*], rgEx480[*Punc-103B: ChR2::YFP* TL5], rgEx481[*Punc-103B:ChR2::YFP* TL2], rgEx502[*Punc-103B:ChR2::YFP, Pgar-3B:ChR2::YFP* TL2], rgEx498[*Punc-103B:ChR2::YFP, Pgar-3B:ChR2::YFP* TL1], rgEx354[*Pgar-3B:ChR2::YFP* TL3], rgEx509[*Pgar-3B:G-CaMP3::SL2:::mDsRed*], rgEx576[*Pgpa-1:G-CaMP3::SL2:::mDsRed*], rgEx546[*Ptry-5:G-CaMP3::SL2:::mDsRed*], rgEx575[*Peat-4:G-CaMP3::SL2:::mDsRed*], rgEx578[*Pocr-2:G-CaMP3:: SL2:::mDsRed*], rgEx560[*Ppkd-2:G-CaMP3::SL2:::mDsRed*], rgEx513[*Pdat-1:G-CaMP3::SL2:::mDsRed*], rgEx567[*Punc-103E:G-CaMP3::SL2:::mDsRed*], rgEx430[*Plev-11:G-CaMP1, Plev-11:mDsRed*], rgEx623 [*Pbas-1:G-CaMP6M::SL2:::mDsRed*], rgEx590[*Pbas-1:CFP*], rgEx624[*Pcat-2:YFP*], rgEx658[*Peat-4:G-CaMP6M::SL2:::mDsRed*], rgEx642[*Pdop-2:G-CaMP6M::SL2:::mDsRed*]. All transgenic strains include *pha-1(lf); lite-1(lf)* and *pha-1(+)* rescue. The strains rgEx629, rgEx630, rgEx628[*Pcat-2:cat-2::SL2:::GFP*], rgEx641[*Pcat-2:cat-2::SL2:::GFP*], rgEx637[*Pdat-1:cat-2::SL2:::GFP*], and rgEx654[*Pdat-1:cat-2::SL2:::GFP*] include *cat-2(e1112);pha-1(lf)* and *pha-1(+)* rescue.

### Assay for 475 nm wavelength-light induced behavior

All males contain a mutation in *lite-1*, which encodes a short wavelength (475 nm, blue) light receptor (*Edwards et al., 2008*). Without this mutation, 475 nm wavelength light activates an avoidance pathway that reduces the efficiency of ChR2 stimulation. In non-integrated lines, brightly expressing L4 males were selected and incubated overnight on plates containing all-*trans* retinol (A.G. Scientific, San Diego, CA). Plates were spread with freshly prepared 50 μM all-*trans* retinol in OP50. For integrated lines, no pre-selection for the brightest males was performed; L4 males were selected and placed on plates containing all-*trans* retinol. The next day, one male at a time was placed on a fresh plate containing retinol. The 475 nm wavelength light was turned on, and the amount of time the males protract their spicules over 60 s was recorded. In males that fully protracted their spicules for 60 s, the 475 nm wavelength light was left on until they ejaculated, retracted their spicules, or 5 min had passed, whichever came first. At least two independently obtained transgenic lines were analyzed for each promoter:ChR2 construct(s).

475 nm wavelength stimulated ejaculation is a rare event. To properly analyze this response, we utilized binary categorical data that allowed us to determine how the number of 'yes' (males did

ejaculate) and 'no' (males did not ejaculate) responses varied between genotypes. We employed Fisher's exact test to analyze our contingency tables. Fisher's exact test is used to determine the p value for small sample sizes. This test is employed on data throughout the manuscript where we wanted to determine if males of a given population could execute a specific behavior that depended only on the male being successful (a yes/no question) and not on the amount of time the male took.

## Plasmid construction

### Cell-specific expression of ChR2

pLR183 contains P*gar-3B:ChR2::YFP* and was constructed as in *Liu et al. (2011)*. pBL228 contains the full length *unc-17* promoter. pBL228 was created using primers attb1unc17pfullfor and attb2unc17p-bac to PCR amplify the full length (10,200 bp) version of the *unc-17* promoter from genomic DNA. The PCR product containing the promoter was recombined with pDG15 (*Reiner et al., 2006*) using Gateway BP clonase II (Life Technologies, Grand Island, NY). pBL228 was recombined with pLR167, containing a Gateway Reading Frame Cassette C.1 in front of *ChR2:YFP* (*Liu et al., 2011*), using Gateway LR clonase II to make pBL238. pJM1 contains a short *unc-17* promoter (P*unc-17S*) driving *ChR2:YFP*. It was constructed using LR clonase to recombine pLR159, containing P*unc-17S*, with pLR167. pLR159 was created using primers attb1unc17pfor and attb2unc17pbac to PCR amplify a short (3678 bp) version of the *unc-17* promoter from genomic DNA and recombining the promoter with pDG15 using BP clonase II. pJM3 contains P*unc-103B:ChR2::YFP* and was constructed using LR clonase to recombine pLR30, which contains Punc-103B (*Reiner et al., 2006*), and pLR167.

### Cell-specific expression of G-CaMP3:SL2::mDsRed

pLR279 [*Gateway Reading Frame Cassette:G-CaMP3::SL2:::mDsRed*] and pLR286 [*Pdat-1:G-CaMP3::SL2:::mDsRed*] were constructed as described in *Correa et al. (2012)*. The following promoter regions were PCR amplified using the primers listed: P*gpa-1* (ATTB1gpa1pro and ATTB2gpa1prodwn) and P*eat-4* (Peat-4(5 kb) and attb1Peat-4R). Following amplification they were recombined with pDG15 (*Reiner et al., 2006*) using BP clonase II to generate pLR26 (P*gpa-1*) and pPC52 (P*eat-4*). pLR279 was recombined using LR clonase II with pLR289 (P*unc-103E*) (*Reiner et al., 2006*), pLR57 (P*gar-3B*) (*Liu et al., 2007*), pBL239 (P*try-5*) (*Smith and Stanfield, 2011*), pLR26 (P*gpa-1*), and pPC52 (P*eat-4*) to generate pLR289, pLR283, pBL240, pBL247, pBL256, and pBL257, respectively.

### Cell-specific expression of G-CaMP6M:SL2::mDsRed

We created a G-CaMP6:SL2::mDsRed plasmid in the following manner. We PCR amplified G-CaMP6 from pGP-CMV-G-CaMP6M (Addgene) (*Chen et al., 2013*) using primers ForUPGCaMP and RevdownGCaMP and everything but G-CaMP3 from pLR279 using primers RevUPGCaMP and FordownGCaMP. We then combined the two PCR products using Clontech's Infusion kit to make pLR305 (Clontech, Mountain View, CA). P*bas-1* was amplified from genomic DNA using fpbas-1 and pbas-1r and recombined with pDG15 to generate pBL262. pLR305 and pBL262 were recombined using Gateway LR clonase II to generate pBL270. Since we found the mDsRed from pBL270 too weak to use as a baseline for fluorescent changes, we made a P*bas-1*:mDsRed construct to co-inject with pBL270. pBL262 and pGW322dsRed were recombined using Gateway LR clonase II to generate pBL279. pPC1, containing P*dop-2* (*Correa et al., 2012*), was recombined with pLR305 to generate pPC88.

### Cloning cat-2

We amplified *cat-2* from genomic DNA using primers fcat2pyl30 and cat2pyl30r. At the same time, pYL30, containing Gateway Reading Frame Cassette A:SL2::GFP, was linearized using primers fpyl30 and pyl30r. The *cat-2* and pYL30 PCR products were combined using Clontech's Infusion kit to make pBL289. P*cat-2* was amplified using primers fpcat-21.4 and pcat-2r and cloned into pDG15 using BP clonase II to make pBL283. pBL283 and pBL262(P*bas-1*) were recombined with pBL289 using LR clonase II to make pBL290 and pBL291, respectively.

### cat-2 and bas-1 expression patterns

pBL262 (P*bas-1*) and pBL283 (P*cat-2*) were recombined with pGW77-C2 and pGW322YFP using LR clonase II to make pBL267 and pBL285.

## Transgenics

Plasmids were co-injected with pBX1, which contains wild-type *pha-1* (*Schnabel and Schnabel, 1990*), into *pha-1(e2123ts);lite-1(ce314)* hermaphrodites unless otherwise indicated. Injected animals were

incubated at 20°C; only animals containing a transgene expressing *pha-1* were able to survive to adulthood. pBL240, pLR289, pLR283, pBL255, pLR160, pJM1, pBL285, pBL267, pPC88, and pLR183 were injected at 50 ng/µl. pJM3 was injected with no other ChR2 expressing plasmid at 150 ng/µl. pJM3 was injected with pJM1 and pLR183 at 50 ng/µl. pBL247, pBL256, and pBL257 were injected at 100 ng/µl. pBL270 was co-injected with pBL279 at concentrations of 5 ng/µl and 0.5 ng/µl, respectively. All G-CaMP lines were screened for their ability to mate. pBL290 and pBL291 were injected at 50 ng/µl into *cat-2(e1112);pha-1(e2123ts)*. At least two transmitting lines were scored for behavior. Transgenic lines containing pBL290 and pBL291 express GFP in the same cells as *cat-2(+)*, allowing us to screen for and remove mosaic animals.

## Integration and mapping

rgEx496[*Punc-17 small:ChR2::YFP, Punc-103B:ChR2::YFP, Pgar-3B:ChR2::YFP*] was integrated using the procedure described in *Mello and Fire (1995)*. Briefly, late L4 to early adult worms were exposed to UV at 340 µW/cm$^2$ after being incubated with 4,5,8-trimethylpsoralen for 15 min. Irradiated worms were then transferred to NGM agar plates plus *Escherichia coli* OP50 and allowed to lay progeny. ~2000 individual F2 hermaphrodites were picked and progeny were screened for complete penetrance of fluorescence. Seven integrated lines were obtained, of which five ectopically ejaculated in response to 475 nm wavelength light. The line with the highest level of ejaculation in response to 475 nm wavelength light was *rgIs6*, used for further studies. *rgIs6* was mapped to LGIV using the procedure described in *Wicks et al. (2001)*.

## Cell ablations

Cell ablations were performed as described in *Bargmann and Avery (1995)*. We used a Spectra-Physics VSL-337ND-S Nitrogen Laser (Mountain View, CA) attached to an Olympus BX51 microscope (Olympus, PA). Ablations were done on 2.5% noble agar pads in 50% water and 50% M9. Ablations at L1 were done using 4 mM NaN$_3$, at L4 using 12 mM NaN$_3$, and in adults using 15 mM NaN$_3$. Non-operated control males were placed on pads+NaN$_3$ for the same amount of time. To remove the somatic gonad plus germ line, we ablated the gonad precursor cells Z1-4 in L1 worms, and to remove only the germ line we ablated the precursor cells Z2-3 (*Kimble and Hirsh, 1979*). The phenotypes we saw in ablated males are not the result of collateral laser damage to the genitalia structures that occurred during the operation. The operation was conducted at the L1 larval stage, and genital structures developed two to three days later during the L4 stage. Any animal that displayed development defects due to anesthetic toxicity or collateral laser damage was not used in the assays.

In adult males that had their spicule tips cut, only the very tip of the spicules was removed with the laser. We expressed CFP in the SPV, SPD, and socket cells to determine that using the laser to remove the spicule tips of adult males resulted in fluorescent cytoplasm leaking out the surgically enlarged spicule opening. Fluorescence in the cell body does not recover (*Figure 9—figure supplement 1*).

Additionally, we discovered that the operation procedure has a significant impact on the refractory period in adult males (*Figure 4—figure supplement 1*). Placing adult males on 2.5% noble agar pads without azide to paralyze the worms is sufficient to significantly reduce the refractory period (*Figure 4—figure supplement 1*). To account for this issue, we picked L4 males in the morning and ablated them when they became adults, ~6 hr later, and allowed the males to recover from the operation overnight. This increased the refractory period length in mock ablated males.

## Mating behavior

The morning of the experiment, 10 µl of a saturated *E. coli* OP50 culture was spotted on an NGM agar plate and allowed to dry. 15 two-days-old *unc-64(lf)* hermaphrodites were transferred to this plate and allowed to incubate for at least 1 hr, to ensure the males would respond to the hermaphrodites. The *E. coli* + hermaphrodites section of the plate was cut out and placed on a microscope slide and transferred to an Olympus BX51 microscope. Recordings were then made using a Hamamatsu ImagEM Electron multiplier (EM) CCD camera (Japan) of the male mating. Each recording was then analyzed for the mating behavior of interest.

### Sperm transfer ranking

Males were given 5 min to mate. If they successfully inserted within 5 min, that time was recorded and divided by 300 s. Males that successfully transferred sperm into the hermaphrodite received no additional score. Males that inserted but did not transfer sperm received a 1. Males that ectopically

ejaculated received a 2, and males that neither inserted nor ejaculated received a 3. Therefore, the males with the lowest scores represent those that were the most successful at transferring sperm to a receptive mate.

### Refractory period

Time was recorded from when the males were placed on the mating lawn to first insertion into the hermaphrodite vulva and from first insert to second insert, the latter being designated the refractory period. The time required for a male to mate the first time was recorded to ensure that manipulations to the male did not adversely affect his overall mating drive and ability.

The insertion times to the 1st and 2nd insert was not recorded for ablated males and their operated controls, with the exception of males ablated when they were adults. Instead, mating was watched on a Leica MZ7.5 dissecting microscope, and hand-held timers were used to determine the refractory period.

### Mating drive

Drive was calculated as the time from when the male was placed on the mating lawn to when the male commenced backward locomotion along a hermaphrodite for the 1st commencement, and from when a male retracted his spicules to when he again started backward locomotion along a suitable mate for the 2nd commencement.

### Time at vulva

The total time a male spent at the hermaphrodite vulva before he inserted was calculated. This measurement records how difficult it is for the male to insert his spicules into easily penetrable hermaphrodites. The time for each individual vulva stay was determined, and then all were added together.

### Vulva passes

The number of times a male's genitals passed the vulva without stopping was recorded. This measurement determines how well the male can sense the vulva region.

### Insertion time

The time from each insertion to retraction was recorded. For the first insert, the total time of insertion is reported. For the 2nd insert, the total insertion time within the first 30 s was recorded, since operated males would insert and retract multiple times.

### Mating potency and progeny count

A male was allowed to mate twice. After each successful mating, the hermaphrodite was removed and placed on an individual plate. Several days later, the mating was scored a success if there were any moving progeny on the plate. If necessary, we counted the moving progeny 2–3 days after spicule insertion (*Guo and Garcia, 2014*).

### Ectopic protraction

Once a virgin male was placed on an agar pad containing hermaphrodites, he was counted as positive for ectopic protraction if his spicules protruded from the cloaca prior to vulva insertion. This phenotype could be seen both when the male was in contact with and separated from a mate.

### Time of sperm transfer

Wild-type males transfer sperm for ~17 s after insertion (*Schindelman et al., 2006*). Videos of *cat-2(e1112)* males showed that while these males can still transfer sperm on occasion, they often do it for a much shorter period of time. To quantify this phenotype, we recorded the time from when sperm was first seen in the uterus to the time when sperm movement could no longer be seen. We considered any male that transferred sperm for ≥10 s to exhibit wild type transfer, and any male that transferred sperm for <10 s or not at all to exhibit the mutant phenotype.

### Rescuing cat-2(lf) defects with exogenous dopamine (DA)

1-day-old adult males were placed on NGM plates with food plus 2 mM DA (Sigma, MO) or water for 4–6 hr. This is below the concentration (15 mM) when exogenous DA starts to paralyze hermaphrodites (*Chase et al., 2004*). Male behavior was then assayed as described previously.

## Measuring males' time to insert on dopamine

Various concentrations of DA were added to the plates and allowed to dry. $H_2O$ was used as a control. Hermaphrodites were added. Males were placed next to the hermaphrodites. The time required for the males to commence mating and insert their spicules was recorded.

## Measuring velocity of backward locomotion during vulva search behavior

1-day-old *let-23(sy1);unc-64(e246);lite-1(ce314)* hermaphrodites, which are vulvaless and paralyzed, were used for this experiment. 25 mM DA or water was added to NGM plates and allowed to dry. Experiment was conducted as in *Liu et al. (2011)*. Measurements were taken from the males traveling 1100 µm–1700 µm along the hermaphrodite.

## Still imaging

Images were taken with either an Olympus BX51 microscope and Hamamatsu ImagEM Electron multiplier (EM) CCD camera or with a Olympus IX81 microscope, csu-xi Yokogawa spinning disk, and Andor iXon EM CCD camera.

## Ca²⁺ imaging

### Free-moving males

An NGM plate was seeded with 10 µl *E. coli* OP50 the day of the experiment. Once the *E. coli* spot was dry, ~10 two-day-old *unc-64(lf);lite-1(lf)* hermaphrodites were added to the plates and returned to the 20°C cultivation temperature used for *C. elegans* for at least 1 hr. ~15 min prior to the start of the experiment, both hermaphrodites and males were moved to room temperature to allow them to adjust. We noticed males do not mate well immediately after being transferred between different temperatures. *unc-64(lf);lite-1(lf)* hermaphrodites move too much for Ca²⁺ imaging. To prevent excessive movement without immediately killing them, a white-hot platinum wire was used to briefly touch their noses, thereby damaging the sensory neurons in the head and preventing basal movement and feeding. This procedure kept hermaphrodites from moving for a few hours before they died. Dead hermaphrodites cannot be used because they do not provide enough body turgor to the males and their flaccid cuticles make it difficult for males to locate the vulva. 2-days-old *unc-64(lf)* hermaphrodites were used for ease of spicule insertion into vulva. Males had a difficult time inserting into 1-day-old *unc-64(lf)* hermaphrodites and, while males easily inserted into 3-days-old *unc-64(lf)* hermaphrodites, males could not maintain insertion due to vulva flaccidity. After hermaphrodite preparation, agar containing the bacteria lawn was cut out and placed on a microscope slide. 1 one-day-old virgin male was then added and the whole assembly was transferred to a fluorescence-equipped Olympus BX52 microscope. Males were manually tracked and visualized using a 10X, 20X, or 40X long working distance objective. The G-CaMP and DsRed signals were recorded simultaneously using a Dual View Simultaneous Imaging Systems with an OI-11-EM filter (Photometrics, Surrey, BC, Canada) and Hamamatsu ImagEM Electron multiplier (EM) CCD camera. The recordings were not started until the male had commenced mating and were stopped when the male retracted his spicules.

### Cover-slip restricted males

To determine Ca²⁺ changes in the gonadal valve in males exposed to oxotremorine S (OxoS), we immobilized virgin 1-day-old males expressing G-CaMP3 in the valve on 5% noble agar pads plus Polybead polystyrene 0.1-µm microspheres (Polysciences, Warrington, PA) (*Kim et al., 2013*) with or without 100 mM OxoS. Males were recorded for 1 min and the standard deviation for the %ΔF/F0 for that one minute was determined (for Ca²⁺ analysis, see below).

## Analysis of Ca²⁺ imaging data

Recordings that managed to capture at least a few frames prior to insertion through intromission and sperm transfer were used for analysis. Using the Hamamatsu SimplePCI (version 6.6.0.0) software, a region of interest (ROI) was centered on the cell(s) or tissues of interest in both the G-CaMP and mDsRed channels. Additional ROIs of the same size were placed on the *E. coli* lawn to record the background fluorescence levels for each image. The mean gray level of each ROI was calculated for each frame, with the ROIs being moved frame-by-frame as necessary to keep the cells or tissues of interest inside the ROI as the male moved. Even though the male was situated at the vulva of an unmoving hermaphrodite throughout the recording, we found that even slight changes of position in the male tail necessitated moving the ROI. We were able to correlate fluorescent changes with

behavior since both the spicules and sperm are auto fluorescent, allowing us to determine spicule insertion and sperm entry into the vulva.

The data were then transferred to Microsoft Excel to calculate the percent change in fluorescence ($\Delta F$) over initial fluorescence (F0) (*Correa et al., 2012*). Photobleaching and other experimental artifacts were accounted for as in *Correa et al. (2012)*. While photobleaching is not significant in G-CaMP over the period of time utilized for the experiment, it is significant for mDsRed. mDsRed fluorescent decay was plotted with respect to time. A decay curve was fit to this line and used to adjust the mDsRed fluorescent levels to remove artifacts caused by bleaching. These adjusted red values were used to determine a green:red fluorescent ratio and the %$\Delta F$/F0 for each cell or tissue of interest was then plotted with respect to time.

### Primers

attb1unc17pfullfor:
GGGGACAAGTTTGTACAAAAAAGCAGGCTGTGTATGGTGGTGGAGCATTCGACAT
attb1unc17pfor: GGGGACAAGTTTGTACAAAAAAGCAGGCTTGCAGACTTTTCCCCAAACTAGC
attb2unc17pbac: GGGGACCACTTTGTACAAGAAAGCTGGGTGACGGGCACGTTGAAGCCCAACT
ATTB1gpa1pro: gggg aca agt ttg tac aaa aaa gca ggc tcatgactttggtatttctttgcagaaactcgcg
ATTB2gpa1prodwn: Ggg gac cac ttt gta caa gaa agc tgg gtctgaagtcttcgaataaatgacattgaataatattg
Fpcat-21.4: GGGGACAAGTTTGTACAAAAAAGCAGGCTGCTCAAAAAGAAAATCCGATTTAAA
TGTCTC
Pcat-2r: GGGGACCACTTTGTACAAGAAAGCTGGGTCTGATCGGTGAGCTGTTTTCGGTGTTG
Peat-4(5 kb): GGGGACAAGTTTGTACAAAAAAGCAGGCTCCCCTCAGGCAAGCACAAAGAA
GAAGAAT
attb1Peat-4R: GGGGACCACTTTGTACAAGAAAGCTGGGTAGGTTTCTGAAAATGATGATGATGAT
GATGG
Fpbas-1: GGGGACAAGTTTGTACAAAAAAGCAGGCTGACTTCCGCGAATCCCCATCC
Pbas-1r: GGGGACCACTTTGTACAAGAAAGCTGGGTTATACCGAACTACTACTGAAAGTTCGAC
fcat2pyl30: TACAAAGTGGTGATCATGAGGTGTCAGAAGGTACAGTAATCC
cat2pyl30r: TTGGGGGATCCTCATTCACATTGTAATCGATATTTTCATCCGATC
fpyl30: ATGAGGATCCCCCAACAGAGTTGTTGATC
pyl30r: GATCACCACTTTGTACAAGAAAGCTGAACG
ForUp GCAMP: ATGGTCGACTCATCACGTCGTAAGTGGAATAAGACAGGTC
RevUp GCAMP: GACCTGTCTTATTCCACTTACGACGTGATGAGTCGACCAT
FordownGCAMP: AAACTACGAAGAGTTTGTACAAATGATGACAGCGAAGTGA
RevdownGCAMP: TCACTTCGCTGTCATCATTTGTACAAACTCTTCGTAGTTT

## Acknowledgements

We would like to thank Xiaoyan Guo, Xin Chen, and Liusou Zhang for discussion and criticism of the experiments and manuscript. We thank Daisy Gualberto for technical assistance. We thank Gillian Stanfield for providing *try-5* plasmids. *C. elegans* strains were provided by the Caenorhabditis Genetic Center. This work was supported by the Howard Hughes Medical Institute.

## Additional information

### Funding

| Funder | Author |
|--------|--------|
| Howard Hughes Medical Institute | L René García, Brigitte LeBoeuf, Changhoon Jee |

The funders had no role in study design, data collection and interpretation, or the decision to submit the work for publication.

### Author contributions

BLB, PC, LRG, Conception and design, Acquisition of data, Analysis and interpretation of data, Drafting or revising the article, Contributed unpublished essential data or reagents; CJ, Conception

and design, Analysis and interpretation of data, Drafting or revising the article, Contributed unpublished essential data or reagents

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
