## [Decision Letter]

Thank you for sending your work entitled “A network of sensory-motor neurons and dopaminergic support cells couples ejaculation and satiation in *C. elegans* males” for consideration at *eLife*. Your article has been favorably evaluated by Eve Marder (Senior editor) and 4 reviewers, one of whom, Peggy Mason, is a member of our Board of Reviewing Editors.

The Reviewing editor and the other reviewers discussed their comments before we reached this decision, and the Reviewing editor has assembled the following comments to help you prepare a revised submission.

This is a comprehensive look at an interesting phenomenon – the post-ejaculatory refractory period – in worms. The authors used many different methods to attack this problem. They come up with an important contribution that answers many of the first order questions while also making clear that the topic is sufficiently rich with complexity to warrant further study.

The reviewers who were unfamiliar with the worm field found reading the manuscript challenging. It was hard to follow. Remember that the length of *eLife* papers is not proscribed but is the length needed to convey the story, no more and no less, so please revise for clarity. Overall, the Introduction and Discussion should place the work within the context of what is known in the *C. elegans* literature. Definitions of refractory period and of satiation should be provided in the Introduction. Please note that the literature in mammals clearly differentiates between refractory period and satiation and there is evidence for different neural circuits controlling them. The situation may be different in worm – the authors seem to convey that the two phenomenons are identical in *C. elegans*.

The figures are very difficult to make out. For example, Figure 1 is very difficult to make out both in terms of color and arrangement. Parts B and C are ok but Parts A and D are not readable and should be extensively revised. I recommend an overhaul of the schematic figures. Where the point to be made is highly anatomical (e.g. which muscles support ejaculation), keep the anatomical relationships intact in the diagram. However, where the points are circuit-related, uncouple the diagrams from anatomical representation. Don't make the cells or structures or muscles anatomically sized or arranged. In other words, move away from the strict anatomy and stylize the topography with some extra space between cellular and muscle elements. Use black and white and patterns: the colors are hard to discern. And please increase the size of the figures. Space is not a particular concern but readability is.

One of the tenets upon which this work rests is that the refractory period is useful because it allows the animal to recover to ejaculate again. But unless we missed it (and this is entirely possible) the authors do not test this. The test is easy. When the refractory period is shortened, is sperm count reduced?

Relating the disorders to pathologies such as eating disorders and drug addiction is an inappropriate stretch. In mammals, satiety in food intake is a completely different event with a completely different neural regulation than mating-induced satiety. All discussion or hints of relations of the current findings in *C. elegans* with those in rodent literature of food intake (or addiction) are suspect, and need to be omitted or substantially toned down. This includes the first paragraph of the Introduction and last paragraph of the Discussion.

It would be worthwhile for the authors to speculate on the relationship between their results and ejaculatory control in mammals. The circuitry controlling ejaculation is well described in rodents and that literature is directly relevant to the studies here, yet not discussed or cited. In addition, there is peripheral modulation of the corpus cavernosum by dopamine and in fact other monoamines, most strikingly serotonin. There may be a general theme here of monoaminergic modulation of sexual motor acts. In comparing the findings to work in mammals, it is also noteworthy that refractory period in worms is defined strictly on the basis of re-initiation of mating. Is this independent of a neural network in the head? If so, that would make it very different from the situation in mammals, where the post ejaculatory refractory period appears to be controlled by networks in hypothalamus, amygdala, and septal nuclei. Some discussion on this would be helpful.

Could the laser ablation phenotypes have been caused by non-specific injury? This is unlikely given the consistency of these results with the rest of the story, but should be commented on.

In the last paragraphs of the Introduction and in the last paragraph of the Discussion the authors comment that the refractory period is likely to aid “recovery” and/or “resetting”. However, they do not comment on what might need to recover or what in the neuromuscular network might need resetting during the recovery period. It would be interesting to know more about their thoughts on this matter (or at least if they found this puzzling).

The refractory period was defined as including both reduced mating drive and less-efficient execution of mating. Defects in gonad-ablated and linker-cell-ablated males suggest the refractory period is subject to regulation, which is the authors' primary conclusion from these experiments. However, it is interesting that failure to transfer sperm leads to a lengthened refractory period in some animals. Why would this be the case? The authors are invited to speculate on this point in the Discussion.

The timing of Ca^2+^ transients in SPD, SPV, and SPC neurons and the gonadal valve suggests these neurons promote sperm transfer. Valve transients can be induced by an ACh agonist, consistent with being regulated by SPC, and show altered kinetics of decay in spicule tips-ablated males. Data from wild-type males suggest a role for SPC in initiation followed by SPD/SPV in continued movement. However, ablation of SPV/SPD slowed the increase of Ca^2+^ in SPC, suggesting the spicule neurons actually promote initiation of release via SPC as well as transfer. These results are apparently conflicting. Why does the SPC peak more rapidly than SPV/SPD? This point seems worth addressing, at least by discussing more fully.

ChR2 stimulation of several groups of neurons were used to evoke “artificial” ejaculation. Curiously, expression in a wide range of cells is necessary for evoked ejaculation. This assay is likely to be useful for subsequent studies in this and other labs. Here, the glutamatergic p.c.s. neuron PCA along with SPC/PCB were shown to promote ejaculation. High PCA Ca^2+^ transients occurred after insertion and decreased at the initiation of sperm transfer. SPV/SPD were not required for artificial ejaculation, suggesting they function upstream of the relevant ChR2-expressing cells; their ablation had no effect on Ca^2+^ in PCA or PCB.

Socket cells of the spicule promote sperm movement into the hermaphrodite and show a rapid increase in transients during mating like those of SPC and the valve. Dopamine synthesis is required in the socket cells rather than in ray neurons for sperm transfer. Socket cell DA also promotes the refractory period. One question that arises based on the presented model is whether exogenous dopamine is sufficient to induce a “refractory period” in the absence of prior mating.

---

## [Author Response]

*The reviewers who were unfamiliar with the worm field found reading the manuscript challenging. It was hard to follow. Remember that the length of* eLife *papers is not proscribed but is the length needed to convey the story, no more and no less, so please revise for clarity. Overall, the Introduction and Discussion should place the work within the context of what is known in the* C. elegans *literature. Definitions of refractory period and of satiation should be provided in the Introduction. Please note that the literature in mammals clearly differentiates between refractory period and satiation and there is evidence for different neural circuits controlling them. The situation may be different in worm – the authors seem to convey that the two phenomenons are identical in* C. elegans*.*

The manuscript was revised for clarity. For example, we added a supplemental figure to Figure 4 to display what effect cutting the spicule tips had on the circuitry. Figure 1 was split into two figures and referenced throughout the manuscript so the reader has images to refer to. Since Figures 1 and 2 provide the reader with the relevant information on the structure of the reproductive circuit in the *C. elegans* male and cell connectivity, we changed the model figure (Figure 10) to a flow chart of the steps of mating behavior and the circuitry we identified to be involved in specific aspects of sperm movement and release and the refractory period.

A paragraph was added to the Introduction explaining how the work builds on what is known in the *C. elegans* literature (paragraph starting: “The well-defined structural components of the nervous system in the *C. elegans* hermaphrodite have facilitated a detailed understanding of how circuits function to produce behaviors …. The tool set used to deconstruct the circuits in hermaphrodites can be applied to study the most complex behavior exhibited by the nematode, male mating”).

We thank the reviewers for alerting us to the different definitions for the word satiation: (1) a feeling of long lasting disgust or fatigue after over engaging in an activity. (2) The termination of an activity after a sense of fullness or satisfaction. We were not using the term in the same way that the rodent sexual behavior researchers currently use the term. The definitions for refractory period and satiation in mammals were added to the Introduction. In the manuscript we originally submitted, we used the term satiation to refer to the successful termination of a mating bout. However, the rodent studies use satiation to refer to the long term inhibition of copulation attempts after ad libidum mating. We do not know if *C. elegans* males ever undergo an extended period of mating inhibition due to excessive mating. To account for this, we removed all references to *C. elegans* satiation. We use satiation in the manuscript to refer to rodents. We added the following paragraph to the Discussion (paragraph starting “Much remains to be discovered concerning the molecular and structural pathways involved in post ejaculatory behavioral activity (51, 92)… since their functional reproductive span is very short”).

*The figures are very difficult to make out. For example,*
Figure 1
*is very difficult to make out both in terms of color and arrangement. Parts B and C are ok but Parts A and D are not readable and should be extensively revised. I recommend an overhaul of the schematic figures. Where the point to be made is highly anatomical (e.g. which muscles support ejaculation), keep the anatomical relationships intact in the diagram. However, where the points are circuit-related, uncouple the diagrams from anatomical representation. Don't make the cells or structures or muscles anatomically sized or arranged. In other words, move away from the strict anatomy and stylize the topography with some extra space between cellular and muscle elements. Use black and white and patterns: The colors are hard to discern. And please increase the size of the figures. Space is not a particular concern but readability is*.

*The figures were revised for clarity. The diagrams in*
Figure 1
*were edited to make them clearer and remove the color.*
Figure 1
*was split into two figures to make the diagrams easier to visualize. What was originally*
Figure 2
*was split into two figures, now labelled*
Figure 3
*and*
Figure 4*, to make the data easier to see. The data in other figures was revised to make the scales on the axis similar for comparison, the text easier to read and the graphs easier to see.*
Figure 10
*was completely revised*.

*One of the tenets upon which this work rests is that the refractory period is useful because it allows the animal to recover to ejaculate again. But unless we missed it (and this is entirely possible) the authors do not test this*. *The test is easy. When the refractory period is shortened, is sperm count reduced?*

To address this question, we performed the following experiment: we allowed wild type males to mate twice to paralyzed hermaphrodites and recorded the refractory period while removing the hermaphrodites to separate plates. We then counted the number of moving progeny, indicating male-sired progeny, two-three days later. Males that did not sire progeny the first time were removed from the study, so that it was not affected by potentially defective males. The results of this experiment are reported in Figure 3.

The conclusion of the experiment was indeed, as the reviewer predicted, the longer the refractory period, the higher the sperm count.

The following paragraph was added to the manuscript in the Results section entitled “Successful copulation is followed by a period of reduced mating drive and ability” (paragraph starting “In mammals, since the amount of sperm decreases in the male reproductive tract after ejaculation, the refractory period may provide an opportunity to re-establish sperm count… We next asked how the various cellular components involved in ejaculation influence the refractory period”).

*Relating the disorders to pathologies such as eating disorders and drug addiction is an inappropriate stretch. In mammals, satiety in food intake is a completely different event with a completely different neural regulation than mating-induced satiety. All discussion or hints of relations of the current findings in* C. elegans *with those in rodent literature of food intake (or addiction) are suspect, and need to be omitted or substantially toned down. This includes the first paragraph of the Introduction and last paragraph of the Discussion.*

We have removed references to food intake and addiction in the Introduction and Discussion.

*It would be worthwhile for the authors to speculate on the relationship between their results and ejaculatory control in mammals. The circuitry controlling ejaculation is well described in rodents and that literature is directly relevant to the studies here, yet not discussed or cited. In addition, there is peripheral modulation of the corpus cavernosum by dopamine and in fact other monoamines, most strikingly serotonin. There may be a general theme here of monoaminergic modulation of sexual motor acts. In comparing the findings to work in mammals, it is also noteworthy that refractory period in worms is defined strictly on the basis of re-initiation of mating. Is this independent of a neural network in the head? If so, that would make it very different from the situation in mammals, where the post ejaculatory refractory period appears to be controlled by networks in hypothalamus, amygdala, and septal nuclei. Some discussion on this would be helpful*.

To address these issues, we added the following paragraph to the Discussion: “In mammals, post-ejaculatory regulation of the refractory period occurs in networks in the hypothalamus, amygdala and spetal nuclei…*C. elegans* ejaculatory circuitry is similarly set up (Figure 10) and allows us to expand the general understanding of circuit control of ejaculatory and post-ejaculatory behaviors.”

To address the role of monoamine regulation of sexual motor acts, we added the following to the paragraph on dopamine: “This role of DA could represent a general theme of monoaminergic modulation of sexual motor acts, as there is also peripheral modulation of the corpus cavernosum in mammals by monoamines, including dopamine and serotonin (2, 35, 19, 82).”

*Could the laser ablation phenotypes have been caused by non-specific injury? This is unlikely given the consistency of these results with the rest of the story, but should be commented on*.

We added the following sentence to the Materials and methods section on cell ablations:

“The phenotypes we saw in ablated males are not the result of collateral laser damage to the genitalia structures that occurred during the operation. The operation was conducted at the L1 larval stage, and genital structures developed two to three days later during the L4 stage. Any animal that displayed development defects due to anesthetic toxicity or collateral laser damage were not used in the assays.”

*In the last paragraphs of the Introduction and in the last paragraph of the Discussion the authors comment that the refractory period is likely to aid “recovery” and/or “resetting”. However, they do not comment on what might need to recover or what in the neuromuscular network might need resetting during the recovery period. It would be interesting to know more about their thoughts on this matter (or at least if they found this puzzling)*.

We found the high level of circuit activity necessary for ejaculation striking and speculate that this activity could require a period of “resetting” or “recovery”. We have added the following paragraph to the Discussion that includes our extended thoughts of what the neuromuscular network needs to recover from (paragraph starting “This raises the question of what does the male mating circuitry need to recover from?... Thus, the high amount of circuit activity required for ejaculation might necessitate a similar period of reduced activity to restore proper circuit function”).

*The refractory period was defined as including both reduced mating drive and less-efficient execution of mating. Defects in gonad-ablated and linker-cell-ablated males suggest the refractory period is subject to regulation, which is the authors' primary conclusion from these experiments. However, it is interesting that failure to transfer sperm leads to a lengthened refractory period in some animals. Why would this be the case? The authors are invited to speculate on this point in the Discussion*.

To address this question, we have added the following to the Discussion (paragraph starting “We propose that the SPV and SPD neurons not only promote sperm initiation but regulate the length of the refractory period… Consequently, lengthening the refractory period would allow time for the males to produce sperm”).

*The timing of Ca*^*2+*^
*transients in SPD, SPV, and SPC neurons and the gonadal valve suggests these neurons promote sperm transfer. Valve transients can be induced by an ACh agonist, consistent with being regulated by SPC, and show altered kinetics of decay in spicule tips-ablated males. Data from wild-type males suggest a role for SPC in initiation followed by SPD/SPV in continued movement. However, ablation of SPV/SPD slowed the increase of Ca*^*2+*^
*in SPC, suggesting the spicule neurons actually promote initiation of release via SPC as well as transfer. These results are apparently conflicting. Why does the SPC peak more rapidly than SPV/SPD? This point seems worth addressing, at least by discussing more fully*.

The SPV/SPD are involved in both steps of ejaculation, initiation, and release. We edited the Discussion to make this point clearer, and added the following paragraph with our model for why the SPC peaks more rapidly than the SPV/SPD, yet is dependent upon the SPV/SPD for its rapid activation (paragraph starting “Interestingly, we found that the rapid increase in SPC Ca^2+^ transients upon insertion is dependent on the SPV and SPD neurons, despite the fact that the SPV and SPD do not reach their peak activity as fast as the SPC…This model accounts for the role of SPV and SPD in SPC activity and initial sperm movement”).

*ChR2 stimulation of several groups of neurons were used to evoke “artificial” ejaculation. Curiously, expression in a wide range of cells is necessary for evoked ejaculation. This assay is likely to be useful for subsequent studies in this and other labs. Here, the glutamatergic p.c.s. neuron PCA along with SPC/PCB were shown to promote ejaculation. High PCA Ca*^*2+*^
*transients occurred after insertion and decreased at the initiation of sperm transfer. SPV/SPD were not required for artificial ejaculation, suggesting they function upstream of the relevant ChR2-expressing cells; their ablation had no effect on Ca*^*2+*^
*in PCA or PCB*.

*Socket cells of the spicule promote sperm movement into the hermaphrodite and show a rapid increase in transients during mating like those of SPC and the valve. Dopamine synthesis is required in the socket cells rather than in ray neurons for sperm transfer. Socket cell DA also promotes the refractory period. One question that arises based on the presented model is whether exogenous dopamine is sufficient to induce a “refractory period” in the absence of prior mating*.

To address this question, we measured the time it took for males to commence mating and insert their spicules on different concentrations of dopamine. The results are reported in Figure 9 and Figure 9—figure supplement 1.

We found that exogenous dopamine will artificially lengthen the time virgin males take to commence mating and insert their spicules. We added the following paragraph to the final Results section entitled “Dopamine released from the spicule support cells promotes sperm release and the refractory period” (paragraph starting “Since endogenous DA extends the period of time between copulations, we asked if exogenous DA is able to extend the time required for a virgin male to mate successfully the 1st time… Thus, exogenous DA is capable of interfering with males’ mating ability”).